# Multi-targeted ¹H/¹⁹F MRI unmasks specific danger patterns for emerging cardiovascular disorders

Ulrich Flögel [1,2,3✉], Sebastian Temme [1,2,3,4], Christoph Jacoby[1,3], Thomas Oerther[5], Petra Keul[6], Vera Flocke[1], Xiaowei Wang [7], Florian Bönner[3], Fabian Nienhaus[3], Karlheinz Peter[7], Jürgen Schrader[2,3], Maria Grandoch[8], Malte Kelm[2,3] & Bodo Levkau[6]

Prediction of the transition from stable to acute coronary syndromes driven by vascular inflammation, thrombosis with subsequent microembolization, and vessel occlusion leading to irreversible myocardial damage is still an unsolved problem. Here, we introduce a multi-targeted and multi-color nanotracer platform technology that simultaneously visualizes evolving danger patterns in the development of progressive coronary inflammation and atherothrombosis prior to spontaneous myocardial infarction in mice. Individual ligand-equipped perfluorocarbon nanoemulsions are used as targeting agents and are differentiated by their specific spectral signatures via implementation of multi chemical shift selective ¹⁹F MRI. Thereby, we are able to identify areas at high risk of and predictive for consecutive development of myocardial infarction, at a time when no conventional parameter indicates any imminent danger. The principle of this multi-targeted approach can easily be adapted to monitor also a variety of other disease entities and constitutes a technology with disease-predictive potential.

[1] Experimental Cardiovascular Imaging, Institute for Molecular Cardiology, Heinrich Heine University, Düsseldorf, Germany. [2] Cardiovascular Research Institute Düsseldorf (CARID), Heinrich Heine University, Düsseldorf, Germany. [3] Department of Cardiology, Pneumology and Angiology, University Hospital Düsseldorf, Düsseldorf, Germany. [4] Department of Anesthesiology, Heinrich Heine University, Düsseldorf, Germany. [5] Bruker BioSpin, Rheinstetten, Germany. [6] Institute of Molecular Medicine III, Heinrich Heine University, Düsseldorf, Germany. [7] Baker Heart and Diabetes Institute, Melbourne, Australia. [8] Department of Pharmacology and Clinical Pharmacology, Heinrich Heine University, Düsseldorf, Germany. ✉email: floegel@uni-duesseldorf.de

Cardiovascular disease is worldwide a major cause of mortality and morbidity[1] and, thus, has made the identification of patients at high risk for myocardial infarction (MI) or stroke a key challenge. A major driver towards vessel occlusion is vascular thromboinflammation, resulting in the activation of pro-thrombotic and pro-inflammatory functions of endothelial cells, which in turn leads to dysregulation of coagulation, complement, platelet activation, and leukocyte recruitment in the microvasculature[2]. For risk assessment and therapeutic decision making, one ideally would like to monitor the entire underlying pathophysiological cascade with early endothelial activation, vascular inflammation, thrombus formation at the arterial wall, and subsequent microembolization, finally culminating into total vessel occlusion and acute MI[3,4]. Simultaneous identification of these interlaced processes would require a multi-level approach for timely visualization of the emerging danger patterns at the culprit site of the coronary circulation, causing transition from stable to acute coronary syndromes with plaque formation over erosion to rupture[5]. Data of the present study demonstrate that this goal can be achieved within a single imaging session using a novel multi-targeted MRI-based approach.

Among the molecular imaging approaches, fluorine ($^{19}$F) MRI has recently emerged as a promising tool for visualizing disease markers that offer information beyond anatomical alterations. $^{19}$F is an intrinsically sensitive nucleus for MRI[6]; there is negligible endogenous $^{19}$F in the body and, thus, no background signal which allows the detection and exact localization of fluorinated materials as 'hotspots' by combined $^1$H/$^{19}$F MRI[7]. In addition, there is a family of compounds, perfluorocarbons, which exhibits a very high fluorine pay-load and that is biochemically and physiologically inert[8]. Perfluorocarbon nanoemulsions (PFCs) are readily ingested by blood monocytes allowing us and others to track their migration into inflammatory foci in vivo[9–12]. Using an 'active targeting' approach by equipping PFCs with an α2-antiplasmin peptide (α2AP) cross-linking to the developing fibrin network by FXIIIa[13,14], also the occurrence of freshly formed thrombi could lately be monitored in vivo[15].

In the present study, we have raised this approach to the next level for simultaneous detection of multiple specific imaging signatures that we have termed cardiovascular danger patterns. We demonstrate that this allows the reliable identification of arterial vasculature at risk for imminent occlusive thrombosis thus predicting the occurrence of MI in the 'HypoE' mouse model[16,17]. This diet-induced model of spontaneous MI closely recapitulates human pathophysiology, including the entire cascade of inflammatory coronary plaque development, atherothrombosis, microembolization, and vessel occlusion with subsequent MI. To visualize the individual steps of thromboinflammation in this model, we developed multi-targeted PFCs with distinct spectral properties for concomitant monitoring of major biological hallmarks of disease progression. For the true simultaneous imaging of multiple PFC signals, we further evolved multi chemical shift selective $^{19}$F MRI allowing the time-saving and artefact-free acquisition of, in principle, any number of $^{19}$F resonance frequencies.

## Results

### Early detection of cardiovascular inflammation prior to loss of endothelial integrity

First, we explored whether our approach can identify inflammatory patterns at exquisitely early stages in this mouse model. For this, repetitive injections of neat (untargeted) perfluorocrown ether emulsions (PFCE) together with late gadolinium (Gd) enhancement (LGE) were employed for longitudinal monitoring of infiltrating immune cells and occurrence of ischemia, respectively (see timeline in Supplementary Fig. 1).

Animals received PFCs two days prior to MRI allowing for adequate $^{19}$F loading of immune cells and their subsequent migration into inflammatory foci. LGE was carried out only after $^{19}$F MRI to avoid alterations of $^{19}$F relaxation times impacting on quantification.

Serial $^{19}$F MRI was performed over 15 days as shown in Fig. 1: As early as day 5 after the initiation of atherogenic diet, $^{19}$F signal was detected at the branching site of right common carotid and subclavian artery; an area known as prone to premature plaque development (Fig. 1a, $^{19}$F superimposed in red over anatomical $^1$H MRI (greyscale)). Follow-up on day 10 revealed increased PFC depositions in the aortic arch and beginning inflammation at the base of the heart (Fig. 1b, c, left). Of note, no signal enhancement was observed at the site of $^{19}$F signal upon Gd application, excluding any loss of membrane integrity at this time (Fig. 1b, c right (LGE) vs. middle (native)). Five days later (day 15), there was a further increase of $^{19}$F signal at the same site indicative of progressing annular inflammation now associated with a congruent LGE pattern (Fig. 1d); the same was observed in the apex consistent with previous reports on MI location[16] in this model (Supplementary Fig. 2). Post mortem 3D high resolution $^1$H/$^{19}$F MRI demonstrated that the $^{19}$F label was co-located with the proximal large coronary arteries, reflecting infiltration of PFC-loaded immune cells into inflammatory hotspots within the coronary ostia and aortic valves as classical predilection sites of atherosclerosis (Fig. 1e, left and middle). Similar time courses were regularly observed in all mice from a cohort of six (see below and Supplementary Figs. 3–7).

Subsequent pathohistological analyses combined with ex vivo MRI always and reliably corroborated the in vivo findings and confirmed (i) PFC deposition at the luminal site of the truncus brachiocephalicus and immune cells in the aortic wall on day 5 (Supplementary Fig. 3), (ii) inflammation at the base of the heart without any signs for MI or necrosis on day 10 (Supplementary Fig. 4), (iii) co-localization of immune cells and myocardial scar tissue at an advanced disease stage on day 15 (Supplementary Fig. 5), and (iv) robust detection of $^{19}$F signals irrespective of the individually different disease progression (Supplementary Fig. 6).

Altogether, these results demonstrated that—in a model exhibiting a continuum from coronary inflammation to vessel occlusion and MI—$^{19}$F MRI allowed an early visualization of inflammatory processes in the coronary circulation prior to any signs of structural damage. Passive distribution of emulsion particles ($\varnothing \geq 100$ nm) within the tissue can be excluded ($\rightarrow$ no LGE), which is an important prerequisite for the specific targeting of epitopes inside the vascular wall. No cardiac $^{19}$F signals whatsoever were observed in mice of the same genotype but without atherogenic diet (Supplementary Fig. 7).

### Multi chemical shift selective imaging for artefact-free detection of different PFCs

Next, we turned to the concurrent detection of ongoing thrombotic processes in this model. To this end, we employed ligands specific for markers of early and chronic thrombosis, such as α2AP[15] and fbn (based on EP-2104R[18]) for targeting of FXIIIa and fibrin, respectively, that we attached to the surface of PFCs with spectral signatures distinct from PFCE: perfluorooctyl bromide (PFOB) and perfluoro-1,3,5-trimethylcyclohexane (PFCH), respectively (Fig. 2b left). Importantly, all targeted PFCs were additionally PEGylated for stealth against uptake by monocytes[15] (Supplementary Fig. 8).

For simultaneous, artefact-free and rapid detection of individual PFCs in a single scan, we developed a multi chemical shift selective imaging (mCSSI) sequence as schematically shown in Fig. 2a. It can be considered as a multi-slice 3D rapid acquisition with relaxation enhancement (RARE) variant, in which slice selection is replaced by

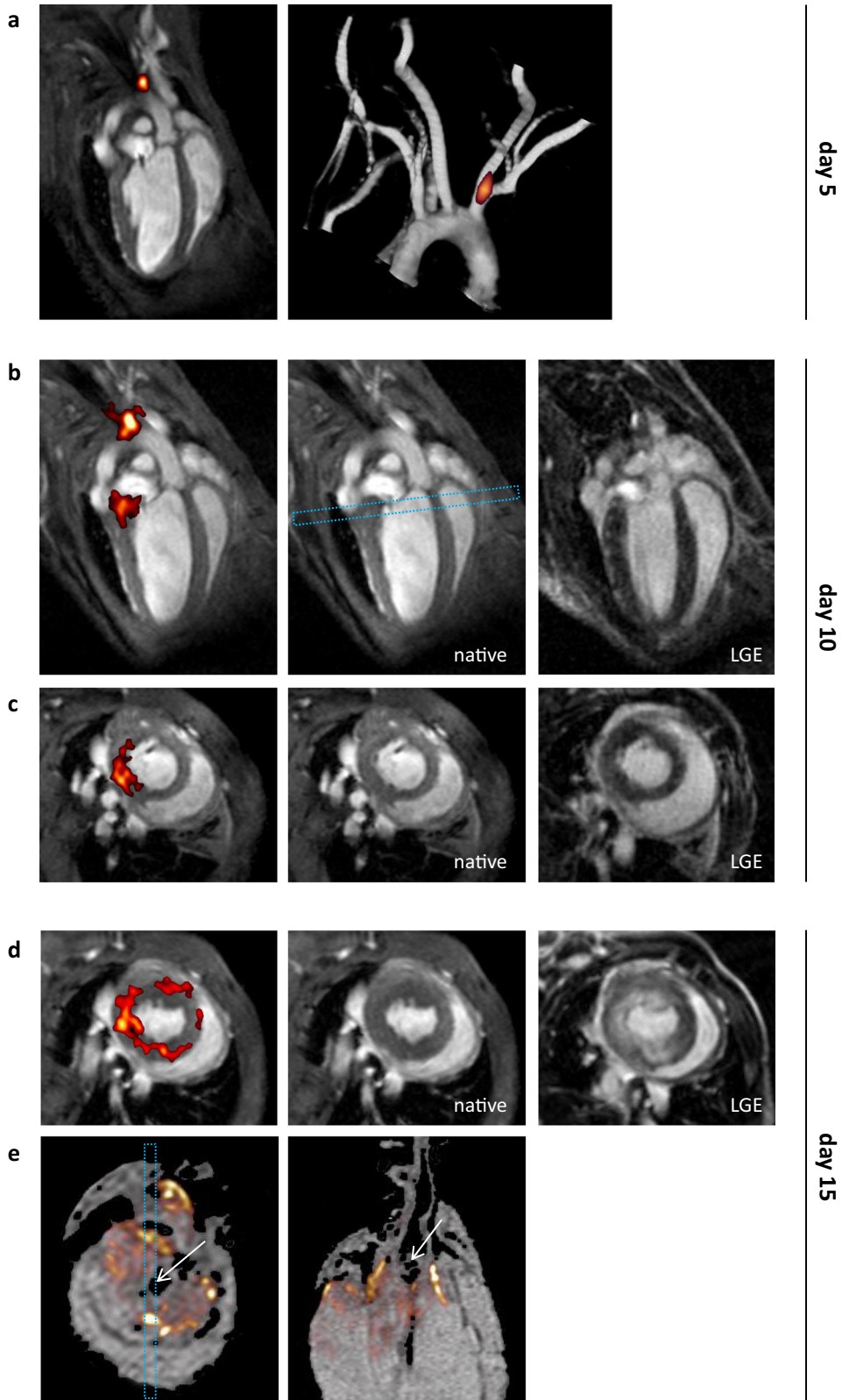

frequency selective excitation. Narrow bandwidth excitation is followed by phase encoding of the third dimension and acquisition of the echo train with selective refocusing pulses. Gradient spoiling destroys remaining transverse magnetization at the end of each cycle and the entire procedure is repeated for the number of excitation frequencies within one repetition time. This approach can be applied in principle to any number of frequencies.

This sequence allowed us to specifically detect all individual $^{19}$F signals without any spill-over of neighboring $^{19}$F peaks and to image the three perfluorocarbons by simultaneous acquisition of five distinct $^{19}$F frequencies (Supplementary Table 1 and indicated by colored arrows in Fig. 2b left). As shown in Fig. 2c, $^{19}$F mCSSI enabled artefact-free detection of both separated and mixed emulsions in the same experiment. For PFCE (red), its

**Fig. 1 Early detection of cardiovascular inflammation and its gradual progression over time by ¹H/¹⁹F MRI.** Tracking of PFC-loaded immune cells in the same mouse 5 days (**a**), 10 days (**b**+**c**) and 15 days (**d**+**e**) after onset of the diet. ¹⁹F MRI data are superimposed in red over anatomical ¹H MRI data (greyscale). **a** First investigation on day 5: Long axis view (left) showing ¹⁹F signal only at the cusp of the aortic arch. The presence of ¹⁹F signal at the branching site of right common carotid and subclavian artery at the brachiocephalic trunk was further confirmed by merging of ¹⁹F and time-of-flight MR angiography data (middle). **b**+**c** Follow-up on day 10 revealed increased PFC deposition in the aortic arch and concomitantly onset of inflammation at the base of the heart (left). The dashed rectangle (middle) indicates the perpendicular view illustrated in c. Upon Gd application, in both views no signal enhancement was observed at the site of inflammation (b+c) right (LGE) vs middle (native). **d** Further check-up on day 15 demonstrated progressing annular infiltration of immune cells at the heart base, now associated with congruent LGE pattern. Thus, tracking of PFC-loaded immune cells early identified myocardial tissue later prone to myocardial ischemia. **e** Post mortem 3D ¹H/¹⁹F MRI at an isotropic resolution of 40 μm revealed that the ¹⁹F label was primarily co-located with the proximal large coronary arteries at the base of the heart (left and middle, white arrows point to the aortic root; dashed rectangle in the short axis view represents spatial orientation of the long axis view), indicating infiltration of PFC-loaded immune cells into inflammatory hotspots within the coronary ostia and aortic valves as classical predilection sites of atherosclerosis.

single frequency was used, while for PFOB (cyan) and PFCH (magenta) two frequencies were excited, resulting in two images for each compound (Fig. 2b). An in-house developed software module permitted separation and image reconstruction for each frequency (Fig. 2b right) as well as summation of the individual free induction decays, in order to increase SNR for PFCs with multiple signals (Fig. 2c). There was only a minor effect of homonuclear J modulation on the utilized frequencies (Supplementary Fig. 9), well-known to occur in the linear PFOB skeleton[19,20]. Thus, a reliable quantification for each signal was provided by ¹⁹F mCSSI (Supplementary Fig. 10).

This flexible multi-color detection of distinct PFCs was then applied for visualization of acute and chronic thrombosis. Individual binding characteristics of $^{α2AP}$PFOB and $^{fbn}$PFCH were verified in ex vivo generated human thrombi (Supplementary Fig. 11) and validated in a murine model of deep venous thrombosis in vivo (Supplementary Fig. 12). As expected, $^{α2AP}$PFOB specifically labeled only freshly developed thrombi, while $^{fbn}$PFCH bound also to thrombi in later stages as robustly detected by ¹⁹F mCSSI (Supplementary Figs. 11b and 12b+c).

**Multi-targeted PFCs for thromboinflammatory states in the heart.** To longitudinally track the interplay of inflammation and thrombosis in the HypoE mouse we applied a mixture of PFCE/$^{α2AP}$PFOB/$^{fbn}$PFCH after onset of the diet. Of note, all emulsions had similar physicochemical properties and comparable ¹⁹F content (Supplementary Fig. 13). Figure 3 illustrates the concurrent monitoring of PFCs targeting inflammation (red), FXIIIa (cyan) and fibrin (magenta) in the same mouse over 15 days, with ¹⁹F mCSSI data superimposed over anatomical ¹H MRI data (greyscale): On day 5, low level cardiac inflammation was found at the heart base (Fig. 3a) indicative for emerging coronary inflammation without any signs of thrombosis and accompanied by normal heart function (Fig. 3b, left ventricular ejection fraction (LVEF) 67.5%). On day 10, inflammation was clearly enhanced and signals for freshly developed thrombi (FXIIIa + fibrin) as well as chronic thrombi (fibrin only) were detected at sites of inflammation that had been identified on day 5 (Fig. 3a,↔3c). This points to pathological tissue alterations without yet functional impairment as LVEF was still preserved (Fig. 3d, LVEF 66.4%). On day 15, however, left ventricular diastolic dilation with severely impaired systolic function had developed (Fig. 3e, LVEF 25.7%) indicative of a functionally relevant MI. These results strongly imply that the initial imaging signals observed within the different fluorine 'channels' have identified regions affected by silent thromboinflammation constituting a danger pattern predictive of a subsequent major MI.

**Premature detection of pulmonary thromboinflammation turns into right heart failure.** Finally, our imaging approach

coincidentally identified pulmonary thromboembolism as a pathological entity hereto unknown in this model. On day 5, multi-targeted ¹⁹F MRI had identified central (hilus) acute (FXIIIa + fibrin) and chronic thrombi (fibrin only) in the lung along with pulmonary areas exhibiting low grade inflammation (red); (Fig. 4a). Interestingly, left and right ventricular (RV) function were yet uncompromised (Fig. 4b; LVEF 70.4%, RVEF 71.1%), that had sharply changed only 5 days later, where massive RV dilation and functional impairment had developed (Fig. 4c; LVEF 54.6%, RVEF 17.8%), a hallmark of pulmonary thromboembolism. As with MI, the emerging signals in the ¹⁹F 'channels' had identified danger patterns indicative for subsequent severe pulmonary thromboembolism (confirmed post mortem by histology, Supplementary Fig. 14).

**Occurrence, distribution, and prognostic value of detected PFC signals.** Quantification of all ¹⁹F MRI experiments ($n = 50$ in 23/17/10 mice on days 5/10/15) for occurrence and distribution of detected danger patterns revealed that the predominantly affected sites were the basal part of the left ventricle containing the coronary ostia/aortic valves and the pulmonary system (Fig. 5a). Linear regression analysis for the ¹⁹F signal at day x and cardiac function on day x+5 (i.e. the investigation five days later) for all follow-up measurements with detectable ¹⁹F patterns ($n = 36$ in 9/17/10 mice on days 5/10/15) revealed an excellent correlation (Fig. 5b). This strongly indicates that simultaneous visualization of different danger patterns by multi-targeted ¹H/¹⁹F MRI can be successfully used to predict further functional impairment.

**Discussion**

Our findings demonstrate that multi-targeted PFCs allow the reliable identification of still silent but prognostically highly relevant thromboinflammation prior to any subsequent ischemic or thromboembolic event. By concurrent detection of several hallmarks of disease progression, it maps the entire transition from beginning coronary inflammation to microembolization, vessel occlusion and MI (Fig. 6). In principle, other disease entities, target epitopes or cell types could be approached by the same technology assuming the design of suitable ligands e.g. against neutrophils and/or platelet-leukocyte aggregates. Thus, the presented approach provides a general and versatile platform for molecular imaging with a diversity that cannot be offered by any other whole-body imaging technique at this time. The mCSSI sequence allows a highly variable spectral readout and simultaneous detection of several different targeting agents taking advantage of the broad chemical shift range for ¹⁹F (~400 vs. ~10 ppm for ¹H). Although multispectral imaging efforts have been undertaken before and are not limited to MRI[21–23], all of the

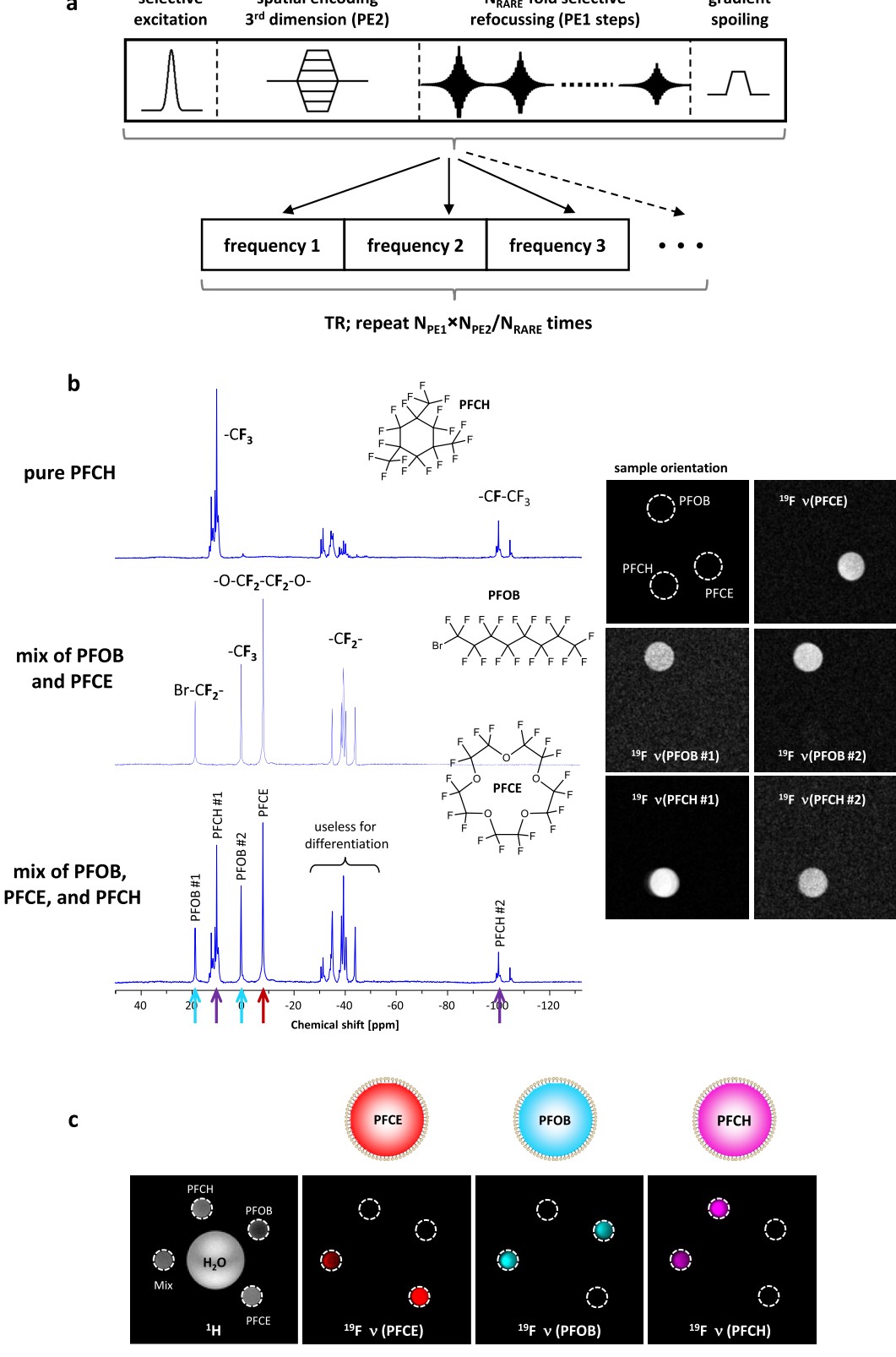

reported attempts[24–27] do not offer the flexibility and specificity of the in situ multi-targeting approach validated in our study.

The developed sequence is in principle applicable to any number of frequencies. Thus, we already identified another PFC, i.e. perfluoro[(tert-butyl)cyclohexane], with a clearly separated resonance frequency (Supplementary Fig. 15 top) as additional 'color'. In proof-of-concept experiments, we verified whether this could be exploited to also visualize another key player in cardiovascular

**Fig. 2 Multi chemical shift selective imaging for artefact-free detection of different PFCs. a** Schematic presentation of the multi chemical shift selective imaging (mCSSI) pulse sequence. It can be considered as a multi-slice 3D rapid acquisition with relaxation enhancement (RARE) variant, in which slice selection is replaced by frequency selective excitation (N$_{PE}$ = number of phase encoding steps, N$_{RARE}$ = RARE factor). Narrow bandwidth excitation is followed by phase encoding of the third dimension and acquisition of the echo train with selective refocussing pulses. Gradient spoiling is required to destroy remaining transverse magnetization at the end of each cycle. This procedure is repeated for the number of excitation frequencies within one repetition time which is similar to TR of the standard RARE sequence. The method provides a complete 3D dataset for each of the excitation frequencies and allows the simultaneous acquisition of several frequencies within one scan. Either individual or (user-defined) sum images can be created from these datasets using dedicated reconstruction software. **b** Left: $^{19}$F MR spectra of pure PFCH (top), a mixture of PFOB + PFCE (middle) and a combination of all three PFCs (bottom). The arrows in the bottom spectrum indicate which signals were used for mCSSI. Chemical structures of PFCH, PFOB, and PFCE are given next to the spectra and signal assignments are indicated in bold. Right: $^{19}$F mCSSI images acquired from the individual resonance frequencies indicated on the left clearly demonstrate artefact-free imaging of the respective five signals. **c** $^{19}$F mCSSI of PFCE, PFCOB and PFCH and a composition of all PFCs (mix). Images are axial cross-sections through five microfuge tubes. The $^{1}$H image (left) shows the position of the samples with a large water phantom in the middle. Adjacent, the reconstructed $^{19}$F images for PFCE (red), PFOB (cyan), and PFCH (magenta) with the last two merged from the individual images #1 and #2, respectively (cf. b right). Note, that in the PFC mixture all three signals were found and that there are no false-positive signals.

thromboinflammation: Since acute arterial thrombosis is usually platelet-rich, we employed as targeting ligand a unique single-chain antibody (scFv[28]), that specifically binds to the activated conformation of GPIIb/IIIa and thus to activated platelets. As demonstrated in Supplementary Fig. 15 middle and bottom, this enabled us to also track the progressive accumulation of activated platelets in diet-exposed mice. Importantly, these experiments corroborated pulmonary thrombosis as up to now unknown pathological characteristic in our mouse model, since we observed substantial platelet aggregation not only in the heart but also in the lung of mice over time (Supplementary Fig. 15 middle and bottom).

The toolbox provided by the present study thus holds the potential to map the transition from stable to unstable coronary syndromes covering the entire cascade from local inflammation, subsequent platelet adhesion, microembolization, and thromboembolism finally culminating in MI. The full encompassment of this process will allow the exact characterization of the ongoing disease state which will not only be useful for mechanistic studies in basic research but also to monitor the effectiveness of novel anti-inflammatory approaches and cellular targets to treat coronary artery disease[29]. Our approach holds as well translational promise for tailored patient therapy: The precise identification in which stage of this pathophysiological sequence the patient is currently in clearly goes beyond exclusive plaque recognition with surrogate parameters of inflammation and/or destabilization of the vascular wall as provided by established clinical imaging techniques. These include adverse coronary plaque characteristics and overall calcified plaque burden[30], microcalcification using NaF[31], enhanced glucose turnover and activated macrophages[32], high intensity plaques in MRI[33], activation of perivascular fat[34], and direct characterization of the arterial wall by invasive approaches[35,36]. Unfortunately, the acuity of inflammation within atherosclerotic plaques not always matches metabolic turnover or degree of plaque activation and exhibits varying specificity depending on the vessel region[32,37]. Furthermore, the diagnostic triad of symptoms, changes in ECG, and elevation of troponin has been shown to reliably identify patients in whom the transition from stable coronary syndromes to acute coronary myocardial damage has already been occurred[38–40]. Similarly, microembolization currently is only detectable at late stage in MRI as microvascular obstruction[41]. Thus, none of these imaging or laboratory-based approaches can fully map the progressive continuum of the entire spectrum of danger patterns indicating the destabilization of arterial culprit lesions with simultaneous detection of vascular inflammation, local atherothrombosis, distal microembolization culminating in thromboembolic events such as acute MI or stroke in a single investigation.

Of course, additional investigations are needed to show that the PFC modifications carried out for specific targeting (PEGylation, ligand equipment) maintain the required safety profile for human use. Several untargeted PFCs have already been evaluated in clinical trials as artificial blood substitutes (e.g. perflubron)[8,42]. Although promising the application of PFCs for this purpose did not really take off, but this was less related to unexpected side effects than that they were not better than the standard of care, i.e. blood[42,43]. Apart from that, anti-PEG antibodies in patients could be a problem for administration of PEGylated contrast agents[44]. On the other hand, PEG is frequently incorporated into drug formulations to improve therapeutic efficacy and in 2020, there were 21 PEGylated drugs approved by the FDA and over 20 others in clinical trials[45]. Nevertheless, much effort is currently undertaken to explore the capacity of other polymers to suppress unspecific nanoparticle uptake as candidates with comparable or superior properties to PEG[46,47]. The modifications made to the nanoparticles may also impact on excretion pathways and biological half-life of the targeted PFCs. It is generally accepted that PFCs are released from the organism by exhalation via the lung which is consistent with the observation that the excretion rate primarily depends on their vapor pressure and permeability through the membrane[48]. However, also PFC transit to the gut has been observed, but this seems to be of minor relevance for humans and non-rodent species[49]. Importantly, there are recent reports about quickly clearing and more sustainable PFC formulations[50,51], which will be helpful for bridging the gap between experimental animal studies and human applications. Nevertheless, doses and biological half-lives have to be specifically tailored for human use, but this is clearly beyond the scope of the present study and has to be adjusted in clinical trials.

In summary, PFCs with unique spectral signatures can be equipped with specific targeting ligands and utilized to simultaneously track loco-regional alterations of different specific danger patterns by multi chemical shift selective $^{19}$F MRI. Using a murine model of atherothrombotic MI mimicking the human situation, we demonstrate that this technology can be applied to serial in vivo imaging of the entire pathophysiological cascade resulting in MI and heart failure and furthermore to identify hereto unknown pathological entities in already well-characterized animal models. Of note, human scanners can readily be equipped with $^{19}$F interfaces[52,53] and improved scanning algorithms[54], coil design[55] and relaxometry of PFCs[56] will facilitate the transmission of preclinical high field findings to the clinical setting. Provided that possible side-effects as discussed above can be overcome, our approach thus offers a potentially translational perspective for identifying patients at risk for myocardial infarction, stroke, or pulmonary thromboembolism but for whom there is no anatomic correlate of risk in conventional diagnosis, allowing for individualized, informed interventions. In addition to more precisely identify the transition from stable to acute vascular disorders, our MRI-based multi-targeted nanotracer platform enables the opportunity to detect early inflammatory changes in plaque formation allowing primary prevention and to monitor effectiveness of novel anti-inflammatory targets and therapies overall enhancing the option

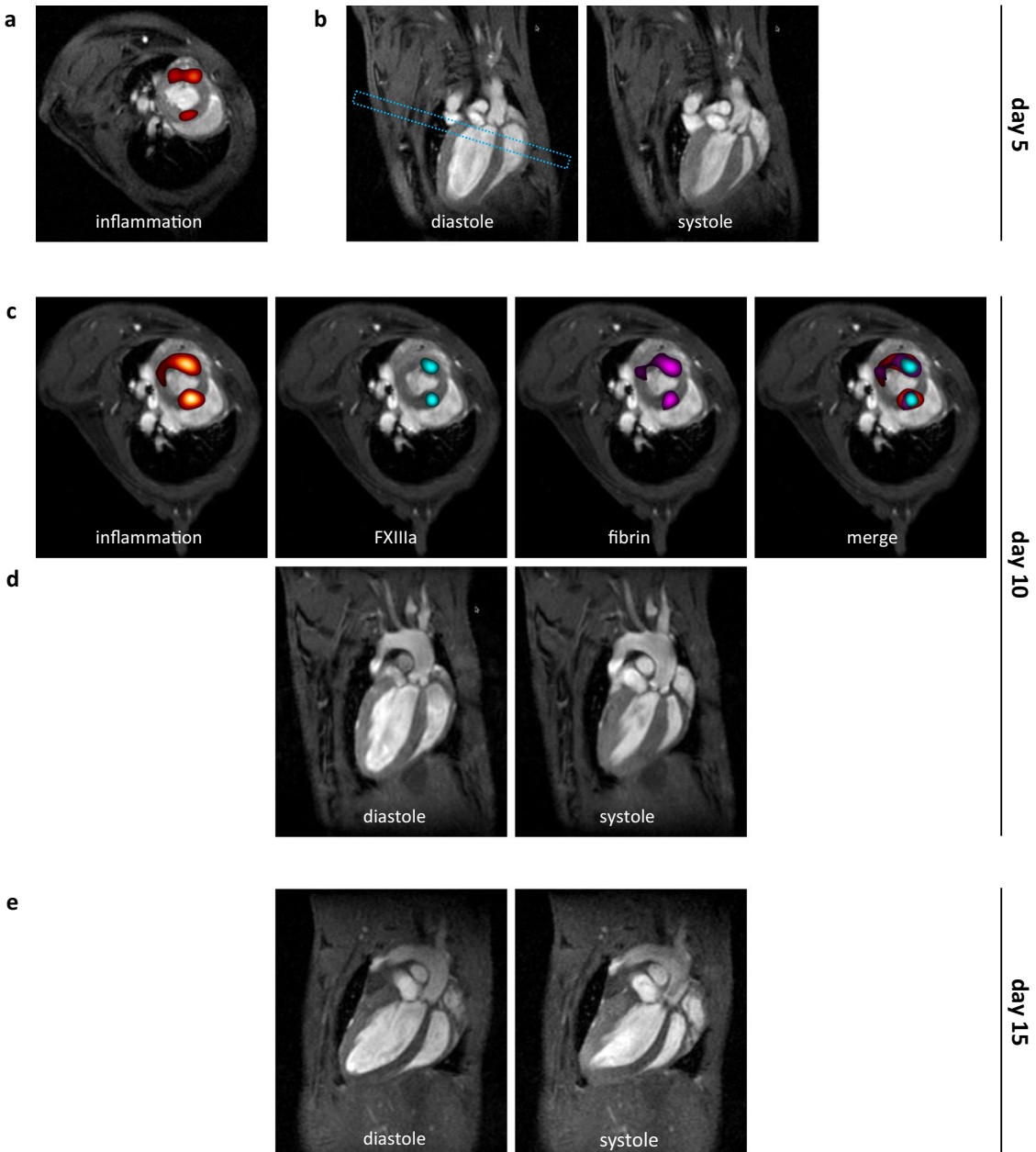

**Fig. 3 Multi-targeted PFCs for thromboinflammatory states.** Concurrent monitoring of PFCs targeting inflammation (red), FXIIIa (cyan) and fibrin (magenta) in the same mouse 5 days (**a**+**b**), 10 days (**c**+**d**) and 15 (**e**) days after onset of the diet. $^{19}$F mCSSI data are superimposed over anatomical $^1$H MRI data (greyscale). First investigation on day 5 indicated **a** low cardiac inflammation at base level and **b** normal function in long axis views in diastole and systole (LVEF 67.5%). The dashed rectangle indicates the perpendicular view shown in a+c. Follow-up on day 10 revealed **c** enhanced infiltration of immune cells and both freshly developed (FXIIIa + fibrin) and persisting thrombosis (fibrin only) at sites of preceding inflammation (cf. a). **d** Despite incipient thrombosis in the left ventricle cardiac function still is preserved at this time (LVEF 66.4%). **e** Further check-up on day 15 depicts diastolic dilation and massively impaired systolic function (LVEF 25.7%).

of individualized precision medicine in disorders of vascular inflammation.

## Methods

### Chemicals and reagents

*Chemicals.* Cholesterol-PEG$_{2000}$-maleimide (Nanocs); Na$_2$HPO$_4$, NaH$_2$PO$_4$, glycerol (CarlRoth GmbH); thrombin, ADP (Sigma Aldrich); E80S (Lipoid GmbH); perfluoro-15-crown-5 ether (PFCE), perfluoro-1,3,5-trimethylcyclohexane (PFCH), perfluorooctyl bromide (PFOB), diblock, i.e. F$_6$H$_{10}$, perfluoro-n-hexyl decane (Abcr chemicals). *Buffer:* Phosphate glycerol buffer (3 mM NaH$_2$PO$_4$, 7 mM Na$_2$HPO$_4$, glycerol 2.5% m/V, pH 7.4), phosphate buffer (PBS, CarlRoth). *Targeting peptides:* Peptides which were derived from α2-antiplasmin (α2AP = Ac–Gly–Asn–Gln–Glu–Gln–Val–Ser–Pro–Leu–Thr–

Leu–Leu–Lys(Cys)–Trp–Lys[13,15]) and a peptide which was modified on the basis of EP2104-R[57] (here called fbn = Tyr–D-Glu–Cys–Hyp–Tyr[3-Cl]–Gly–Leu–Cys–Tyr–Ile–Gln–Gly–Gly–Gly–Cys) were utilized for targeting. As control for α2AP, we used a peptide where the glutamic acid at position Q3 which is cross-linked to the fibrin network was replaced by alanine. The control peptide for fbn (FAM–Ser–Asp–Gly–Gly–Als–Asn–Leu–Gly–Gly–Ile–Glu–Gly–Gly–Gly–Cys) was generated by replacing and randomizing all of the amino acids which are responsible for fibrin binding[58]. All peptides were commercially manufactured (Genaxxon Bioscience) and fbn and its control were equipped with an additional C-terminal GGG-Cys tag and a carboxy-fluorescein at the N-terminus to enable fluorescence imaging. The FXIIIa specific peptide α2AP and its control contained an additional Cys- on the ε-amino group of K13 and the carboxyfluorescein was conjugated to K15. Delivered lyophilized peptides were suspended in phosphate glycerol buffer and stored at −20 °C.

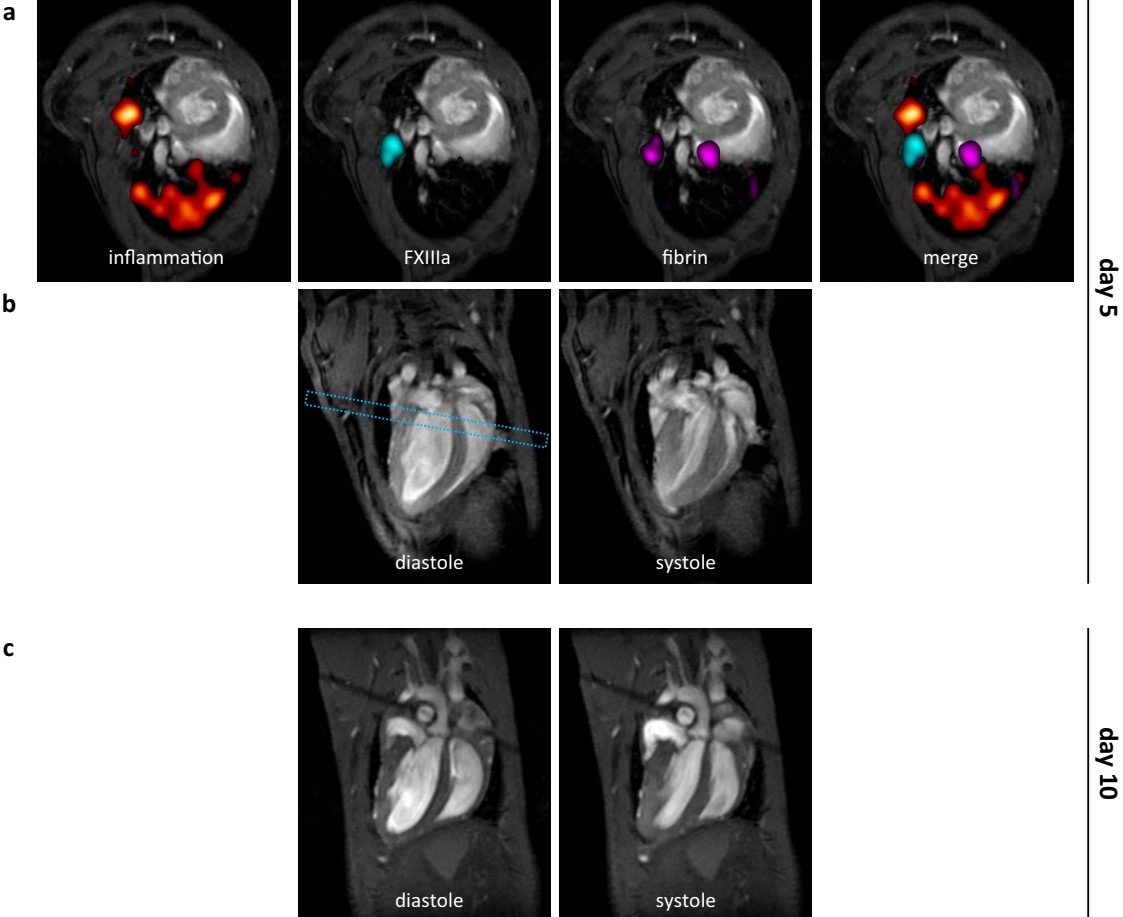

**Fig. 4 Premature detection of pulmonary inflammation and thrombosis turns into right heart failure.** Multi-targeted PFCs for detection of inflammation (red), FXIIIa (cyan) and fibrin (magenta) in the same mouse 5 days (a+b) and 10 days (c) after onset of the diet. **a** ¹⁹F mCSSI data are superimposed over anatomical ¹H MRI data (greyscale) and first investigation on day 5 indicated differentially affected tissue sites mainly in pulmonary areas with **b** normal function in long axis views in diastole and systole (LVEF 70.4%, RVEF 71.1%). The dashed rectangle indicates the perpendicular view shown in a. **c** Follow-up on day 10 revealed that preceding pulmonary thromboinflammation (a) leads to severe right ventricular impairment (LVEF 54.6%, RVEF 17.8%).

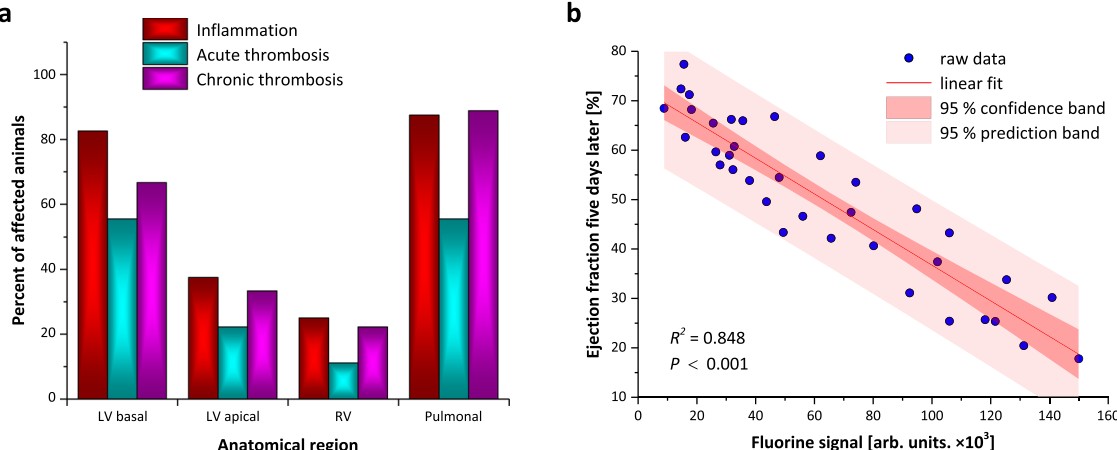

**Fig. 5 Occurrence, distribution, and prognostic value of detected ¹⁹F signals. a** Occurrence and distribution of detected danger patterns are expressed as percentage of animals affected by inflammation (red), acute (cyan) and chronic (magenta) thrombosis in different anatomical regions ($n = 50$ in 23/17/10 mice on days 5/10/15; LV, RV = left and right ventricle). **b** Linear regression analysis for detectable ¹⁹F signal at day x and cardiac function at day x+5, i.e. the follow-up investigation five days later ($n = 36$ in 9/17/10 mice on days 5/10/15, adjusted $R^2 = 0.84811$, $P = 2.926 \times 10^{-15}$). The measure of center for the error band is given by the regression line of the linear fit with $y = -0.0004x + 72.816$ and the $P$ value was determined from ANOVA sum of squares. Source data are provided as a source data file.

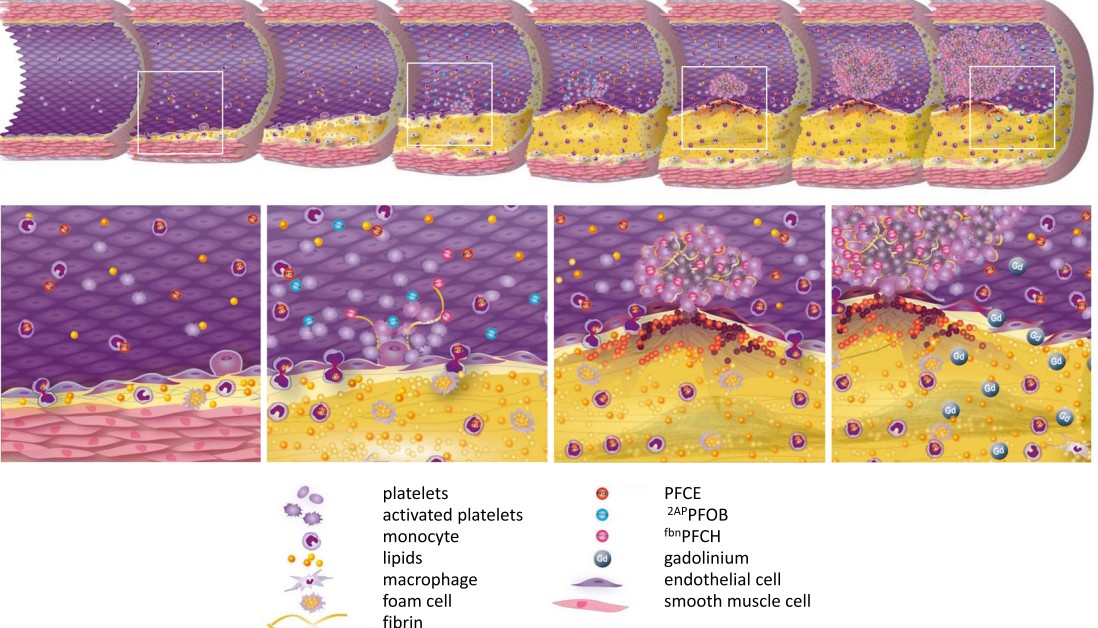

**Fig. 6 Unmasking of emerging danger patterns by multi-targeted PFCs.** Schematic drawing of the pathophysiological cascade from beginning coronary inflammation to atherothrombosis and vessel occlusion and the respective epitopes identified by multi-color $^{19}$F MRI. Top: Overview of the atherosclerotic continuum from coronary inflammation to occlusion in HypoE mice upon exposure to atherogenic high-fat diet. White rectangles indicate the magnified areas shown below. Bottom: 1$^{st}$ column: Uptake of injected PFCE droplets by circulating monocytes and early infiltration of $^{19}$F-loaded cells into inflamed endothelium; 2$^{nd}$ column: acute thrombosis leads to (i) incorporation of $^{\alpha 2AP}$PFOB into the developing fibrin network *via* cross-linking by FXIIIa and (ii) binding of $^{fbn}$PFCH to fibrin strands; 3$^{rd}$ column: Advanced thrombi are labeled by $^{fbn}$PFCHs only with no detectable signal for $^{\alpha 2AP}$PFOB due to lack of FXIIIa. Acute and advanced thrombosis are accompanied by progressing infiltration of PFCE-loaded monocytes into surrounding tissue; 4$^{th}$ column: Vessel occlusion results in ischemia accompanied by loss of endothelial integrity and late gadolinium enhancement.

## Preparation and characterization of perfluorocarbon nanoemulsions (PFCs)

*Preparation.* PFCs which contained perfluoro-15-crown-5 ether (PFCE), perfluorooctyl bromide (PFOB) or perfluoro-1,3,6-trimethylcyclohexane (PFCH) were prepared by microfluidization with E80S lipids as emulsifier using a laboratory processor (LV1, Microfluidics Corp.) operating at 1000 bar. PFCH and PFOB emulsions were further stabilized by adding (equimolar to the lipid content) a semifluorinated fluorocarbon/hydrocarbon diblock[59]. In brief, 2.4% (w/w) E80S was dissolved in 10 mM phosphate glycerol buffer. Subsequently, either PFCE (20% w/w), PFCH (40% w/w) or PFOB (40% w/w) was added to the lipid suspension. For PFCH and PFOB 35 mM diblock was additionally added. The suspensions were preemulsified by high shear mixing (Ultra Turrax TP 18/10) and processed with the LV1 at 1000 bar and for 5 cycles. PFCs were autoclaved (121 °C, 1 bar, 30 min), aliquoted in glass vials, and stored at 4 °C under light protection.

*Characterization.* PFCs were characterized by dynamic light scattering (DLS) using a Nanotrac instrument (Microtrac) to determine the hydrodynamic diameter, the polydispersity index (PDI) and the ζ potential. For this, 20 µl of PFCs were diluted in 1 ml Milli-Q water and analyzed for 90 s. These measurements were repeated for 10 times for calculation of mean values. To gain information on several preparations, the mean values and standard deviations of individual particle size measurements were averaged. For determination of the $^{19}$F amount, 10 µl of PFCE, PFOB, PFCH were diluted with 20 µl H$_2$O and filled into a 200 µl reaction tube. Additionally, a 30 µl mixture of PFCs (10 µl each) was prepared. Subsequently, the reaction tubes were fixed to a 2 ml water tube and subjected to $^1$H/$^{19}$F MRI (see below).

## Functionalization of PFCs

PFCs were equipped with peptide ligands using the sterol-based post-insertion technique (SPIT)[15,60]. For this, peptides were coupled via free cysteines to cholesterol-PEG$_{2000}$-maleimide anchors exploiting the formation of a stable thioether after reaction of the cysteine SH-group with the maleimide of the anchor. For conjugation, 280 µg of the peptide (suspended in phosphate glycerol buffer) was added to 300 µl of cholesterol-PEG$_{2000}$-maleimide (1 µg/µl, suspended in phosphate glycerol buffer) and the volume was adjusted to 650 µl with phosphate glycerol buffer. The solution was incubated overnight at 22 °C under constant stirring. On the next day, 650 µl of the cholesterol-PEG$_{2000}$-peptide conjugate was mixed with 650 µl of preformed PFCs and incubated at 22 °C for 3 h which led to the spontaneous insertion of the cholesterol conjugate into the lipid layer of the PFCs[15,60]. For targeting of activated platelets, PFCs were equipped with a unique human single-chain antibody (scFv) as previously described[28]. Functionalized PFCs were stored at 4 °C under light protection.

## Ethics

Animal experiments were performed in accordance with the European Union guidelines described in the directive 2010/63/EU and were approved by North Rhine Westphalian State Agency for Nature, Environment and Consumer Protection (LANUV = Landesamt für Natur, Umwelt und Verbraucherschutz Nordrhein-Westfalen), Germany. In vitro studies with blood samples from healthy humans were conducted in accordance with the declaration of Helsinki and approved by Ethics Committee of the Heinrich Heine University, Düsseldorf, Germany. All healthy volunteers gave written informed consent to the collection and use of blood samples in this study.

## Animals

Mice expressing a hypomorphic mutant form of ApoE (ApoE-R$^{h/h}$) and lacking the scavenger receptor class B type I (SR-BI$^{−/−}$) were bred and housed with a 12L/12D cycle (light switch 6:00/18:00) at an ambient temperature of approximately 22 ± 2 °C and a humidity of 55% at the central animal facility of the Heinrich Heine University Düsseldorf, Germany. Mice were fed a standard chow diet and received tap water ad libidum. For induction of the disease model[16,17], mice were exposed to a high fat/high cholesterol diet (7.5% cocoa butter, 15.8% fat, 1.25% cholesterol, 0.5% sodium cholate; Altromin). For visualization of thromboinflammatory processes by $^{19}$F MRI, mice received an intravenous bolus injection of PFCs (3 mM/kg BW) 48 h prior to MRI to ensure appropriate PFC deposition at the sites of interest[9].

A total of 33 mice (males and females, 25–30 g body weight; 8–10 weeks of age) were used in this study. Twenty-three of them were subjected to the main protocol illustrated in Supplementary Fig. 1. All mice received baseline $^1$H MRI, PFC injections on day 3 after onset of the diet and $^1$H/$^{19}$F MRI on day 5. Thereafter, 6 mice were sacrificed for 3D high resolution $^1$H/$^{19}$F MRI and histology. For follow-up, 17 mice were subjected to PFC injection on day 8 and $^1$H/$^{19}$F MRI on day 10, whereupon again 6 mice were sacrificed for post mortem analyses. Afterwards, one mouse died spontaneously, so that 10 mice received the final PFC injection on day 13 and $^1$H/$^{19}$F MRI on day 15. Thereafter, the remaining mice were sacrificed for 3D $^1$H/$^{19}$F MRI and pathohistological analyses. Additional 4 mice were used for the explorative study of the fourth color with $^{scFv}$PFCs (Supplementary Fig. 15) and further 6 mice served as control and were not subjected to the diet but received PFC injections as indicated above (Supplementary Fig. 7).

## Magnetic resonance imaging

*General.* Data were recorded at a Bruker AVANCE$^{III}$ 9.4T wide bore NMR spectrometer driven by ParaVision 5.1 (Bruker) and operating at frequencies of 400.21 MHz for $^1$H and 376.54 MHz for $^{19}$F measurements[9,15]. Images were acquired using the Bruker microimaging unit Micro 2.5 with actively shielded gradient sets (1.5 T/m) and a

25 mm birdcage resonator (Bruker) tunable to both [1]H and [19]F. Mice were anaesthetized with 1.5% isoflurane and kept at 37 °C. To prevent dehydration of eye surfaces and mucous membranes, the anesthesia gas mixture was moisturized by passing the mixture through a gas-washing bottle filled with water. The front-paws and the left hind-paw were attached to ECG electrodes (Klear-Trace; CAS Medical Systems) and respiration was monitored by means of a pneumatic pillow positioned at the animal's back. Vital functions were acquired by a M1025 system (SA Instruments) and used to synchronize data acquisition with cardiac and respiratory motion if necessary. After acquisition of initial [1]H pilot scans, the resonator was immediately tuned to [19]F to screen the entire thorax above the liver up to the carotid bifurcation for [19]F hotspots. Thereafter, the resonator was tuned back to [1]H to acquire the corresponding anatomical information for detected [19]F signals and functional analysis. For superimposing the images of both nuclei, the 'hot iron' color look-up table provided by ParaVision was applied to [19]F images. To fade out the background noise from [19]F images a constant threshold was applied to [19]F data and signals originating from the liver were masked for sake of clarity (cf. Supplementary Fig. 7 for a representative example of a mouse 10 days after onset of the diet illustrating a comparable [19]F signal intensity in inflamed myocardium as compared to the liver as major site of PFC deposition). For subsequent intraperitoneal Gd contrast agent application (bolus of 0.2 mmol Gd-DTPA per kg body weight), the animal handling system was shortly removed from the magnet and afterwards re-inserted with exactly the same positioning. The entire scanning protocol took around 80 min and was well tolerated by all animals, which recovered within 2 min from anesthesia.

For *functional and morphometric analysis*, high resolution images of mouse hearts were acquired using an ECG- and respiratory-gated segmented fast gradient echo cine sequence with steady state precession (FISP). A flip angle (FA) of 15°, echo time (TE) of 1.2 ms, 128 segments and a repetition time (TR) of about 6-8 ms (depending on the heart rate) were used to acquire 16 frames per heart cycle with a field of view (FOV), $30 \times 30$ mm$^2$; matrix size after zero filling (MS), $256 \times 256$; slice thickness (ST), 1 mm; number of averages (NA), 2; acquisition time (TAcq) per slice for one cine loop, ~1 min. In cases of inadequate ECG recording, images were acquired with a retrospectively gated fast low angle shot sequence (IntragateFLASH, Bruker) using a FA 10°, TE 1.26 ms, TR 5.82 ms, FOV $30 \times 30$ mm$^2$, MS $256 \times 256$, TAcq 1.5 min. For retrospective gating, a navigator slice was placed near the base of the heart, where the signal is primarily modulated by the periodically changing atrial and aortic blood volume[61]. Routinely, 8–10 contiguous short axis slices were required for complete coverage of the LV. Longitudinal slices orientated perpendicular to the atrio-ventricular level served to ensure appropriate scanning of the entire heart from base to apex. For evaluation of functional parameters, ventricular demarcations in end-diastole and -systole were manually drawn with the ParaVision Region-of-Interest (ROI) tool.

*MR angiography (MRA)* of aortic arch and carotid arteries was carried out using a flow-compensated 2D time-of-flight (TOF) FLASH sequence (FA 80°, TE 2.2 ms, TR 12.3 ms, NA 4, TAcq 4.4 min, FOV $30 \times 30$ mm$^2$, MS $384 \times 384$)[62,63]. In z-direction the FOV was set to cover the entire vessel system from the aortic root up to carotid bifurcation (40 overlapping slices with a slice thickness (ST) of 0.4 mm and an interslice distance of 0.25 mm resulting in a package extent of 10.15 mm). For 3D visualization, data were imported into Amira 4.0 (Mercury Computer Systems) and resampled to isotropic voxel size using a Lanzcos filter. Afterwards, [1]H MRA data were visualized in greyscale by texture-based volume rendering (VRT). For overlay, anatomical corresponding [19]F MRI data (see below) were rendered by application of the 'glow' colormap.

**[19]F MRI.** For the initial experiments with PFCE only, [19]F RARE images were essentially recorded as previously described[9,64–66] using the following parameters: RARE factor 32, TR 2500 ms; TE 4.37 ms, MS $128 \times 128$, ST 2 mm, averages 256, TAcq 21.3 min. For multi-color experiments, the *mCSSI pulse sequence* schematically shown in Fig. 2a was developed. It can be considered as a multi-slice 3D RARE variant in which slice selection is replaced by frequency selective excitation. This allows for simultaneous imaging of molecules with complex spectra (Fig. 2b) and/or different substances. Narrow bandwidth excitation is followed by phase encoding of the third dimension and the echo train acquisition with selective refocusing pulses. Gradient spoiling is required to destroy remaining transverse magnetization at the end of each cycle. This procedure is repeated for the number of excitation frequencies within one repetition time which is similar to TR of the standard RARE sequence. Reconstruction is straightforwardly performed like for a standard multislice RARE dataset, in which slice selection is replaced by different narrow band excitation loops, and provides either a complete image set for each frequency or—to omit a manual calculation step afterwards—the user can also choose to directly create sum images from individual frequencies of a PFC with multiple resonances (such as PFOB) to improve SNR. For this, dedicated reconstruction software was developed in-house based on the LabVIEW package (National Instruments). Multi-color [19]F datasets were recorded with this sequence using the following parameters: TR 2500 ms, RARE factor 32, TE 4.49 ms; bandwidths 1370 and 1610 Hz for excitation and refocussing Gaussian pulses, respectively; MS $64 \times 64$, ST 2 mm, NA 512; TAcq 21.3 min. The frequencies for excitation/refocussing were adjusted to the resonances of PFOB, PFCE, PFCH, and PFTBCH as listed in Supplementary Table 1. All chemical shifts relative to the CF$_3$ resonance of PFOB were derived from non-selective spectroscopic measurements. The standard deviation of the average frequencies never exceeded 40 Hz and were well below the bandwidth of the Lorentzian-shaped lines. For [19]F mCSSI, it is therefore adequate to define only one absolute excitation frequency and to assume constant chemical shifts for the other resonances.

For *post mortem high resolutions measurements and subsequent histology*, hearts were excised and fixed in a 1.5 ml Eppendorf cap filled with paraformaldehyde. [19]F data were acquired with a 3D RARE sequence (RARE factor 32, FOV $22 \times 16 \times 16$ mm$^3$, MS $384 \times 256 \times 256$, TE 21.4 ms, TR 2500 ms, NA 72, TAcq 25.36 h), while anatomical reference data were recorded with a 3D FISP sequence from the same FOV with the following parameters: TE 3.4 ms, TR, 6.8 ms, MS $512 \times 384 \times 256$, NA 24, TAcq 51 min. After MRI measurements, serial 5 μm sections of the entire heart were stained with hematoxylin and eosin (H&E).

MRI data were imported into 3D visualization software Amira and resampled to isotropic voxel size using a Lanzcos filter. [1]H MRI data were visualized in greyscale by VRT adapting the lower and upper intensity thresholds for rendering in a way that the surrounding fluid got invisible. For overlay, the 'glow' colormap was used to render the morphologic corresponding [19]F data. After coregistration, data were analyzed in multi-planar view for exact anatomical localization of the [19]F label in the hearts. For confirmation of the [19]F label to be localized at the luminal site of the aorta (Supplemental Fig. 3), the vessel wall was manually segmented and its surface calculated with unconstrained smoothing. For overlay, semi-transparent surface views of the wall were generated and anatomic corresponding [19]F data were rendered as described above.

*Phantom experiments.* To prove reliable quantification of the [19]F mCSSI signals, phantom measurements were performed using the same PFCs as in in vivo experiments. PFCs were filled into 200 μl microfuge tubes and fixed in the resonator with in-house constructed specimen holder. MRI data were acquired with same parameters given above and afterwards SNR was determined for each individual excitation frequency (see below).

*[19]F MR image analysis.* [19]F MRI data were analyzed using an in-house developed software module based on the LabVIEW package (National Instruments) which allowed (i) an overlay on anatomical corresponding [1]H MRI data, (ii) an easy separation and image reconstruction for each frequency, and (iii) also a summation of the individual free induction decays to increase SNR for PFCs with multiple signals. For quantification of the [19]F signal, ROIs were drawn around the area of interest, whereas background ROIs of similar geometry were placed outside the samples. The signal-to-noise ratio was calculated from the mean of the background-corrected [19]F signal divided by the standard deviation of the noise. For phantom measurements, single frequency as well as sum images were quantified from identical ROIs covering the cross sections of the individual PFC-filled vials. For a more detailed description of the [19]F MRI approach, acquisition parameters, and quantification procedures, please refer to references[9,64].

## Cellular uptake of neat, PEGylated and PEGylated functionalized PFCs

*Preparation of Atto$^{647}$PFCs.* Atto$^{647}$ labeled PFCs were prepared as described above. In brief 100 μg of Atto$^{647}$-DPPE (DPPE = 1,2-dipalmitoyl-sn-glycero-3-phosphoethanolamine; ATTO-TEC GmbH) was resuspended in phosphate-glycerol buffer containing solubilized E80S lipids and stirred for 30 min at room temperature. Subsequently, PFCs were added and a pre-emulsion was generated by high-shear mixing. The crude emulsion was subjected to five cycles of micro-fluidization (LV1, Microfluidics) at 1000 bar.

*PEGylation and functionalization of PFCs.* PEGylated PFCs were generated by incubation of 100 μl of Atto$^{647}$PFCs with 100 μl of cholesterol-PEG$_{2000}$-CH3 (45 μl PEG + 55 μl buffer). To generate ligand-functionalized PFCs, Fbn or α2AP were conjugated to cholesterol-PEG$_{2000}$-maleimide and then inserted into the lipid surface of the PFCs by SPIT. To this end, 12.2 μl Fbn or 13.4 μl α2AP (2.5 μg/μl each) were incubated with 39.2 μl cholesterol-PEG$_{2000}$-maleimide. The volume was adjusted to 100 μl with phosphate glycerol buffer and incubated overnight at 20 °C under constant motion. On the next day, 100 μl of the Fbn/α2AP conjugate and additional 12 μl of cholesterol-PEG$_{2000}$-CH$_3$ (10 μg/μl) were added to 100 μl of Atto$^{647}$PFCs and incubated for 3 h at 20 °C.

*Cellular uptake of PFCs by RAW macrophages and murine monocytes.* RAW macrophages were detached from the culture flask by strong shaking of the flask. Approximately $5 \times 10^5$ cells were taken up in 500 μl of culture medium and incubated with 25 μl of PFCs at 37 °C for 20 min. Subsequently, cells were transferred into a FACS tube filled with 1 ml of ice cold FACS buffer (PBS, 2% FBS, 1% EDTA). Cells were washed twice with FACS buffer, finally resuspended in 500 μl FACS buffer with DAPI (1 μg/ml; to label dead cells) and analyzed on a FACS Canto II (BD Biosciences). For analysis of PFC uptake by *murine monocytes*, blood was obtained by venous puncture of heparinized mice. Subsequently, the withdrawn blood (~100 μl) was subjected to hypotonic erythrocyte lysis (3 ml) for 5 min at room temperature, pelleted by centrifugation and washed with ice cold FACS buffer. Incubation with PFCs was again carried out for 20 min at 37 °C. Afterward, cells were washed and labeled with antibodies against CD11b (CD11b-PE, clone M1/17; BD Biosciences) and Ly6G (Ly6G-FITC, clone 1A8, BioLegend) for 30 min at 4 °C, washed twice with FACS buffer, stained with DAPI and analyzed by flow cytometry. Monocytes were identified as CD11b$^+$/SSC$^{low}$ and Ly6G negative. Mean fluorescence signals of RAW macrophages and murine monocytes were recorded in the APC channel and fluorescence signals of cells which were not exposed to PFCs

were subtracted from the data of cells incubated with neat, PEGylated, and PEGylated functionalized PFCs. Please refer to Supplementary Fig. 16 for the gating scheme used to analyse RAW macrophages and murine monocytes.

**In vitro generation and labeling of thrombi**. Platelet rich plasma (PRP) was obtained by venous puncture from healthy volunteers and immediately centrifuged at 200 g for 10 min. The white PRP on the top of the sample was carefully withdrawn, separated in 1 ml aliquots and stored at −20 °C. For generation of acute thrombi, 1 ml of the PRP was melted on ice and mixed with 100 μl of thrombin (5 U/ml) and 10 μl of ADP (1 mM). Then 100 μl samples were transferred to wells of a round-bottom 96-well plate and incubated for 5 min. Next, $^{\alpha 2AP}$PFOB and/or $^{fbn}$PFCH were added and incubated for additional 90 min. Thrombi were removed and washed three times with 5 ml of PBS and analyzed by IVIS (see below) or $^{19}$F MRI. To obtain subacute thrombi, PRP was activated with thrombin and ADP (as described above) and 100 μl samples were incubated in a round-bottom 96-well plate for 90 min. Formed thrombi were removed from the well, washed with PBS and incubated with fbn or ctr peptide or $^{fbn}$PFCH/$^{\alpha 2AP}$PFOB (or the respective controls). To determine the binding of PFCs to chronic thrombi, thrombi were stored in PBS for 24 h, three days and five days at 4 °C and then incubated with the targeting probes, intensively washed with PBS and analyzed by IVIS.

**Imaging of in vitro generated thrombi**. Thrombi labeled with fbn/α2AP-peptides or $^{fbn/\alpha 2AP}$PFCs were spotted on a glass plate and the fluorescence signal of the carboxyfluorescein was determined by an IVIS Lumina II imaging system (Perkin Elmer). For quantification, ROIs were drawn around the thrombus-associated fluorescence signals and the background-corrected mean values were determined. For $^{19}$F MRI, thrombi were placed in 200 μl reaction tubes filled with PBS which were fixed at a 2 mL water tube.

**Induction of deep venous thrombi in vivo**. Mice were kept under anesthesia with 1.5% isoflurane and buprenorphin (0.3 mg/kg) was injected s.c. 30 min prior to surgery for analgesia. A median laparotomy was performed, and the inferior vena cava was exposed at the anatomical level of both kidneys. Subsequently, a filter paper (1 × 2 mm$^2$) soaked with 10% FeCl$_3$ was placed on the top of the vessel and incubated for 4 min[15]. To prevent any leakage of the FeCl$_3$ to the surrounding tissue, two stretches of parafilm were placed on both sides of the vessel. After removal of the filter paper the vessel was washed with 0.9% NaCl to remove residual FeCl$_3$. PFCs (3 mmol/kg body weight) were injected into the tail vein approximately 5 min prior to or 24 h post thrombus induction. Combined $^1$H/$^{19}$F MRI scans were performed 24 h after PFC injection.

**Data reporting**. No statistical methods were used to predetermine sample size. The experiments were not randomized and the investigators were not blinded during experiments and outcome assessment. Due to the highly variable phenotype of the used animal model, each animal was analyzed on its own and included into the study.

**Statistics and reproducibility**. Unless otherwise indicated, all values are given as mean ± standard deviation (SD). Statistical and regression analysis was performed using OriginPro 2016 (Originlab Corporation, Wellesley Hills, USA). Data were tested for Gaussian distribution using D'Agostino and Pearson omnibus normality test. Statistical significance was assessed by Student's two-sided t test compared to the respective control group. Histology was carried out in 22 mice and the representative findings displayed in the supplement were confirmed in at least 6 independent experiments.

**Reporting summary**. Further information on research design is available in the Nature Research Reporting Summary linked to this article.

## Data availability

All data supporting the findings of this study are available within the article and its supplementary material. Any remaining raw data are available from the corresponding author upon reasonable request. Source data are provided with this paper.

## Code availability

All in-house developed LABView modules used for raw data reconstruction are available on request.

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

## Acknowledgements
We thank Bodo Steckel for excellent technical assistance. This work was supported by the Deutsche Forschungsgemeinschaft (SFB 1116 to U.F., M.G., J.S., M.K., B.L.; TRR 259 to U.F., F.B., M.G., M.K.; FL303/6-1 to U.F.; INST 208/764-1 FUGG to U.F., TE1209/1-1 to S.T.) and the European Commission (MSCA-ITN-2019 'NOVA-MRI' to U.F., J.S.; MSCA-RISE-2019 'PRISAR2' to U.F.).

## Author contributions
U.F. and B.L. conceived of the main concept and guided the project; S.T., F.B., and M.K. contributed to the study design; U.F., S.T, C.J., and V.F. performed MRI experiments and analyzed the data; S.T. prepared and characterized PFC emulsions; C.J. developed dedicated software modules for MRI data reconstruction and analysis, T.O. developed the mCSSI sequence; P.K. and B.L. carried out histopathological analysis; X.W. and K.P. developed and provided the scFA; F.N. contributed to MRI experiments; F.B., J.S., M.G., and M.K. helped with critical advice and discussion. U.F. and M.G. drafted the manuscript and all authors discussed the results and contributed to the writing of the paper.

## Funding

## Competing interests
The authors declare no competing interests.

## Additional information

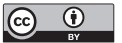

