## [Peer Review File · Nature Communications]

Reviewers' comments:

Reviewer #1 (Remarks to the Author):

The manuscript by Fogel et al describes the implementation and validation of a novel, a multispectral method based on ^{19}F MRI for the longitudinal detection of inflammation, and thrombus (acute and chronic) in the heart and vasculature of a murine model of myocardial infarction. The manuscript is well written and the experimental design very clear. A few specific comments follow.

Specific comments

1. 'However, it requires radioactivity, a nearby cyclotron and additional radiation exposure for anatomical CT.' This statement, referring to the PET radiotracer ^{18}F -NaF is partially incorrect. Unlike other PET tracer labeled with short half-life radioisotopes, ^{18}F -NaF does not require a nearby cyclotron, but can be easily purchased at a very reasonable cost from commercial radiopharmacies. The statement that ^{18}F -NaF 'requires' radioactivity is also misleading (the tracer 'delivers' radioactivity, or 'is' radioactive). Last but not least, using integrated PET and MR scanners can mitigate concerns about additional exposure to ionizing radiation coming from CT.
2. Can the author give more details on the mCSSI reconstruction?
3. In Figure 3, why are not ^{19}F images shown for day 15?
4. In the Discussion 'signals originating from the liver were masked for sake of clarity.' : can the author provide also some images in the supplement to show the extent of liver uptake?

Reviewer #2 (Remarks to the Author):

In their paper "Multi-Targeted $^1\text{H}/^{19}\text{F}$ MRI Unmasks Specific Danger Patterns for Emerging Cardiovascular Disease" Flögel et al present data on previously reported targeted ^{19}F loaded liposomes for the differential diagnosis of CVD. The paper is interesting and extends on the work this group has pioneered over many years. The paper is well written with the multi chemical shift selective imaging (mCSSI) sequences developed the most interesting part of the paper as it allows true multicolour imaging of various ^{18}F nanoparticle formulations.

Main comments:

- The paper would highly benefit from input from a clinician well versed with the current status of the field. Many well-advanced technologies, some already in clinical practice, are missing in the introduction and discussion e.g. non-radioactive CT Ca scores with very low X-ray doses are a very good predictor of CVD. A Ca score of 0 excludes any coronary plaques. Several invasive diagnostic technologies should be mentioned and discussed e.g. infrared scanning Infraredx (recently published in The Lancet that this can identify vulnerable plaques) or Nirtek (Australia).

- LGE is sometimes challenging because of the fast clearance of this small compound. A time course of the maximum uptake in the damaged heart tissue in this model would be important to make sure the correct time points for imaging after injection are selected. Alternatively, Gd labelled larger permeability markers (e.g. dextran) could be used.

- The control data of ¹⁹F tracers injected into mice without an atherogenic diet needs to be shown and included in the supplemental data.

- Acute arterial thrombosis is typically fibrin poor and platelet-rich so it is unclear why the presented targeting motifs have been selected especially given that the group has just published an anti-GPIIb/IIIa targeted ¹⁹F particle system. It would be important to include this imaging in the assessment of inflammation and thrombosis in this mouse model.

- The maleimide conjugation chemistry is problematic in vivo as the bond is reversible. Why can't the targeting peptide be manufactured directly with the Cholesterol-PEG2000 anchor? This would give more confidence about the in vivo stability. Alternatively, biotin tagging as published by the group recently for the GPIIb/IIIa construct should be considered.

Minor comments:

- The first sentence states that anatomical markers of diseases are absent in men. Should that read "Mankind"? CVD is more prevalent in the female population. This should be rephrased in close consultation with a clinician working in the CVD field.

Reviewer #3 (Remarks to the Author):

Multi-Targeted ¹H/¹⁹F MRI Unmasks Specific Danger Patterns for Emerging Cardiovascular Disease

The authors present an interesting preclinical study with outstanding imaging technique performed at 9.4 T. At this high field magnetic strength ¹⁹Fluorine signal, spectral resolution, and spatial resolution are very high. However, the clinical situation at 3 T, would not as efficacious.

Comments:

1. While ¹⁹F background in mice and man is negligible, the prevalence of activated macrophages and smoldering inflammation throughout the human body is very high, especially on a Western Diet or with a myriad of inflammation associated disease. The uptake of PFC particles by splenic, bone marrow, and blood phagocytic cells (Macs) is always high if saturated as in this study, and migration of activated macrophages to sites of inflammation is constantly occurring. Now seen as a snapshot. In the mouse, the early forms of atherosclerosis are induced on a high fat diet and the PFC particles serve as a phagocytosed marker to see that activity. The situation in man is more complicated.
2. The consideration of simultaneous fluorine spectral analysis is not itself new and has been discussed for the last 20 years and demonstrated by other labs long ago, mostly by implanting the PFC particles in different location, perhaps after stem cell uptake ex-vivo with two types of PFOB and PFCE.
3. The images of the three PFCs spectra are in the same location of the heart. The authors provide no data regarding the differential uptake of PFC particles through phagocytosis pathways and cell surface targeting. The general endocytic pathway may account for 80% of the signal versus targeting, despite pegylation over time. Perhaps by blocking phagocyte endocytosis with Cytochalasin D ex vivo, one might help in rooting out this question over 2 or more day time frame. From the data shown, there is no evidence that targeting matters. All the spectral overlay in the same cardiac region, and may be there by nonspecific uptake and trafficking of activated Macs from blood, spleen, or marrow.
4. Thromboembolism is not uncommon with PFC particles, and the cause of the PE reported may be due to aggregation of PFC particles getting entrapped in the lungs. This is what is likely seen. The

authors report sudden deaths with two mice, and that happens when particles are injected too fast, their cell surface homing chemistries interact, or other nonspecific aggregation during the injection process. It happens.

5. The dose of the PFC particles administered is very high, when considered on a human scaled use. How much of this dose is needed to saturate the macrophage targets and how much is required to overcome rodent-related hepatobiliary clearance. All rodents rapidly clear particles, even up to 300nm or more, through the biliary into the gut. Humans, do not. Some of the PFC used have very long biological half-lives and will be persistent. These doses of PFC particles have been associated with adverse flu-like symptoms in man, during the days of artificial blood substitute work, and probably do in mice too. They just don't talk.

6. Since PFC particles will persist at the inflammation sites, the serial uptake of PFC particles is the sum of signal carry-over from the previous injection plus the migration of current inflammation to the site. Like a pulse chase experiment. The residual ¹⁹F signal prior to each serial injection needs to be provided to understand the incremental changes.

7. The uptake of particles, including PFC, may actually incite macrophages to polarize to an M1-like phenotype and this may also contribute to the migration flux to inflamed sites. Perhaps more consideration of this is warranted.

Overall, while the study is interesting from a strict MRI community sense, the translational impact is low, detracting from the general readership. The innovation of multispectral imaging is old and not strongly proven as a differentiating molecular imaging technique in vivo in this experimental design. Again, all of the signal goes to a well-documented location for plaque development and inflammation proportional to the degree of disease in this model, which has been shown by this and others before.

RESPONSE LETTER

We thank all reviewers for the careful evaluation of our manuscript and their constructive criticism. Their suggestions and comments have helped to significantly improve our manuscript. We have addressed the various issues point by point as outlined below.

Response to Reviewer #1

The manuscript by Flögel et al describes the implementation and validation of a novel, a multispectral method based on ^{19}F MRI for the longitudinal detection of inflammation, and thrombus (acute and chronic) in the heart and vasculature of a murine model of myocardial infarction. The manuscript is well written and the experimental design very clear. A few specific comments follow.

Thank you for this positive evaluation!

Specific comments

1. ‘However, it requires radioactivity, a nearby cyclotron and additional radiation exposure for anatomical CT.’ This statement, referring to the PET radiotracer ^{18}F -NaF is partially incorrect. Unlike other PET tracer labelled with short half-life radioisotopes, ^{18}F -NaF does not require a nearby cyclotron, but can be easily purchased at a very reasonable cost from commercial radiopharmacies. The statement that ^{18}F -NaF ‘requires’ radioactivity is also misleading (the tracer ‘delivers’ radioactivity, or ‘is’ radioactive). Last but not least, using integrated PET and MR scanners can mitigate concerns about additional exposure to ionizing radiation coming from CT.

Thank you for making us aware of these inconsistencies. We agree and have completely rewritten this first part of the introduction omitting any statements about radioactivity and radiation exposure (page 2, 2nd paragraph):

Cardiovascular disease is world-wide a major cause of mortality and morbidity¹ and, thus, has made the identification of patients at high risk for myocardial infarction (MI) or stroke a key challenge. A major driver towards vessel occlusion is vascular thromboinflammation resulting in the loss of the normal anti-thrombotic and anti-inflammatory functions of endothelial cells, which in turn leads to dysregulation of coagulation, complement, platelet activation, and leukocyte recruitment in the microvasculature². For risk assessment and therapeutic decision making, one ideally would like to monitor the entire underlying pathophysiological cascade with early endothelial activation, vascular inflammation, thrombus formation at the arterial wall, and subsequent microembolization, finally culminating into total vessel occlusion and acute MI³. Simultaneous visualization of these processes would require a multi-level approach to image these danger patterns at the culprit site of the coronary circulation causing transition from stable to acute coronary syndromes. Data of the present study demonstrate that this goal can be achieved within a single imaging session using a novel multi-targeted MRI-based approach.

Please note: The bibliography to the references herein are only found in the revised manuscript and not within this Response Letter.

2. Can the author give more details on the mCSSI reconstruction?

We are happy to comply: Reconstruction is straightforwardly performed like for a standard multislice RARE dataset, in which slice selection is replaced by different narrow band excitation loops, and provides either a complete image set for each individual frequency or – to omit a manual calculation step afterwards – the user can also choose to directly sum individual frequency images of a PFC with multiple resonances (such as PFOB) to improve SNR. This information has now been included into the Method section dealing with the mCSSI sequence (page 15, line 18 ff).

3. In Figure 3, why are not ^{19}F images shown for day 15?

At the final examination of the multicolor experiments, animals were already seriously affected by the atherogenic diet with a phenotype close to heart failure (see also the severely impaired EFs given in Figures 3 and 4). As a consequence, at this time inflammatory atherothrombosis already expanded over the entire thorax and ^{19}F images for each target looked pretty similar losing their discriminative power (see below). Thus, we felt these images to be less helpful and did not include them into Figures 3 and 4.

4. In the Discussion ‘signals originating from the liver were masked for sake of clarity.’ : can the author provide also some images in the supplement to show the extent of liver uptake?

Again, we are happy to comply and added a representative example of a mouse 10 days after onset of the diet illustrating a comparable ^{19}F signal intensity in inflamed myocardium as compared to the liver as major site of PFC deposition (Supplemental Figure S7 bottom). Furthermore, we included in this Figure a quantification of the peak ^{19}F signal in inflamed

myocardium *vs.* reticuloendothelial accumulation in the liver of mice exposed for 10 days to the diet or normal chow (n = 6 each). Detectable ¹⁹F signals in the heart were restricted to mice subjected to the atherogenic diet and were in the entire cohort comparable to the liver signal intensity (see screenshot of this Figure below).

Response to Reviewer #2

In their paper “Multi-Targeted ¹H/¹⁹F MRI Unmasks Specific Danger Patterns for Emerging Cardiovascular Disease” Flögel et al present data on previously reported targeted ¹⁹F loaded liposomes for the differential diagnosis of CVD. The paper is interesting and extends on the work this group has pioneered over many years. The paper is well written with the multi chemical shift selective imaging (mCSSI) sequences developed the most interesting part of the paper as it allows true multicolour imaging of various ¹⁹F nanoparticle formulations.

Thank you for this positive evaluation!

Main comments:

- The paper would highly benefit from input from a clinician well versed with the current status of the field. Many well-advanced technologies, some already in clinical practice, are missing in the introduction and discussion e.g. non-radioactive CT Ca scores with very low X-ray doses are a very good predictor of CVD. A Ca score of 0 excludes any coronary plaques. Several invasive diagnostic technologies should be mentioned and discussed e.g. infrared scanning Infraredx (recently published in The Lancet that this can identify vulnerable plaques) or Nirtek (Australia).

We agree and as suggested we have rewritten large parts of Introduction and Discussion with valuable input from our colleagues of the Clinic of Cardiology. In the Discussion, we now mention several clinical scores derived from CT, PET, MRI as well as invasive optical techniques and discuss in detail the added value of our approach (page 8, line 22 ff):

The toolbox provided by the present study thus holds the potential to map the transition from stable to unstable coronary syndromes covering the entire cascade from local inflammation, subsequent platelet adhesion, microembolization, and thromboembolism finally culminating in MI. (...) The precise identification in which stage of this pathophysiological sequence the patient is currently in clearly goes beyond exclusive plaque recognition with surrogate parameters of inflammation and/or destabilization of the vascular wall as provided by established clinical imaging techniques. These include adverse coronary plaque characteristics and overall calcified plaque burden²⁹, microcalcification using NaF³⁰, enhanced glucose turnover and activated macrophages³¹, high intensity plaques in MRI³², activation of perivascular fat³³, and direct characterization of the arterial wall by invasive approaches^{34,35}. Unfortunately, the acuity of inflammation within atherosclerotic plaques not always matches metabolic turnover or degree of plaque activation and exhibits varying specificity depending on the vessel region^{31,36}. Furthermore, the diagnostic triad of symptoms, changes in ECG, and elevation of troponin has been shown to reliably identify patients in whom the transition from stable coronary syndromes to acute coronary myocardial damage has already been occurred³⁷⁻³⁹. Similarly, microembolization currently is only detectable at late stage in MRI as microvascular obstruction⁴⁰. Thus, none of these imaging or laboratory-based approaches can fully map the progressive continuum of the entire spectrum of danger patterns indicating the destabilization of arterial culprit lesions with simultaneous detection of vascular inflammation, local atherothrombosis, distal microembolization culminating in thromboembolic events such as acute MI or stroke in a single investigation.

Please note: The bibliography to the references herein are only found in the revised manuscript and not within this Response Letter.

- LGE is sometimes challenging because of the fast clearance of this small compound. A time course of the maximum uptake in the damaged heart tissue in this model would be important to make sure the correct time points for imaging after injection are selected. Alternatively, Gd labelled larger permeability markers (e.g. dextran) could be used.

We agree that the routinely used Gd-based contrast agents are quickly cleared via the kidneys. However, since we are using LGE in mice for several years¹⁻⁸, we feel pretty confident to have chosen the adequate time points for imaging after contrast agent injection. In our hands, LGE starts to develop in infarcted murine myocardium already a few minutes after intraperitoneal injection reaching its maximal contrast to healthy myocardium after 15-20 minutes, which is still weakly detectable after 60 minutes and completely gone after 2 hrs at the latest (see graphic below). Since we applied in the present mouse our routine LGE protocols (including injected volume of contrast agent, planning and localization of scans, imaging timing and protocols) to screen for infarcted myocardium and assess infarct size, we feel confident about the validity of the presented data.

- The control data of ¹⁹F tracers injected into mice without an atherogenic diet needs to be shown and included in the supplemental data.

We are happy to comply and show the requested data in Supplemental Figure S7 top. To further confirm reliable and comparable PFC injections in control mice and mice exposed to the

atherogenic diet, we also determined the ^{19}F content in the liver as major site of nanoparticle deposition. Quantification of the ^{19}F MR images of in mice exposed for 10 days to the diet or normal chow demonstrated that detectable ^{19}F signals in the heart were restricted to mice subjected to the atherogenic diet, while passive PFC deposition in the liver was equal in both groups (n=6 each, see screenshot of this Figure below).

- Acute arterial thrombosis is typically fibrin poor and platelet-rich so it is unclear why the presented targeting motifs have been selected especially given that the group has just published an anti-GPIIb/IIIa targeted ^{19}F particle system. It would be important to include this imaging in the assessment of inflammation and thrombosis in this mouse model.

This point is well taken! As suggested, we carried out additional experiments to verify whether the targeting system mentioned by the reviewer is also applicable to the mouse model used in the present study. However, instead of repeating all the multicolor experiments with this additional targeting motif, we combined this with introducing a fourth 'color' for multiplex ^{19}F MRI. For this, we identified per-fluoro[(tert-butyl)cyclohexane] as useful showing a clearly separated resonance frequency from the other PFCs (Supplementary Fig. S15 top). Equipping the latter with the single-chain antibody scFv⁹, that specifically binds to the activated conformation of GPIIb/IIIa enabled us indeed to track also the progressive accumulation of

activated platelets in heart and lung of mice exposed to the diet. As can be seen in Supplementary Fig. S15 middle and bottom, we already observed weak ^{19}F signals in the heart and/or the lungs on day 5 after onset of the diet which strongly increased until day 10 and day 15 (see also screenshot of the Figure below). Of note, quantification of the ^{19}F MR images revealed that animals 1+2 displayed strong ^{19}F signals in the heart whereas in animals 3+4, the ^{19}F signals in the lungs outweighed those in the heart. Importantly, these experiments corroborated pulmonary thrombosis as up to now unknown pathological characteristic in our mouse model. These data strongly support the hypothesis that our approach has the capability to visualize and differentiate pathological developments in complex animal models which more closely mimics the convoluted situation in human patients. The results of these additional experiments are discussed at page 8, line 18 ff.

Concerning the first part of the reviewer's remark: The reviewer is absolutely right that arterial thrombi have lower fibrin content but a higher amount of platelets, compared to venous thrombi.

Nevertheless, we have chosen fibrin as target because it is found in both venous and arterial thrombi as well as in acute and more advanced thrombi in relatively high concentrations¹⁰. Since the principle aim of the current approach was to provide a proof-of-concept that multi-targeted, multi-spectral ¹⁹F MRI can be used for simultaneous imaging and discrimination of different thromboinflammatory danger patterns, we combined to this end the targeting of very early thrombi (α 2AP+fbn), advanced thrombi (fbn only) and infiltrating immune cells (neut PFCs).

- The maleimide conjugation chemistry is problematic in vivo as the bond is reversible. Why can't the targeting peptide be manufactured directly with the Cholesterol-PEG2000 anchor? This would give more confidence about the in vivo stability. Alternatively, biotin tagging as published by the group recently for the GPbII/IIIa construct should be considered.

We fully agree with this reviewer that stability of the targeting construct under *in vivo* conditions is of crucial importance. The main reason for choosing the cysteine-maleimide conjugation is that this system is very flexible and enables the easy conjugation of different ligands to the same scaffold. For this we used a peptide-PEG-cholesterol conjugate based on three building blocks: (i) polyethyleneglycol (PEG) (bb1), (ii) cholesterol (bb2) and (iii) the peptide (bb3). To this end, we obtained the cholesterol-PEG₂₀₀₀-maleimide (a conjugate of bb1 and bb2) and the peptides (bb3) from commercial suppliers and, finally, we just conjugated bb1/bb2 to bb3.

The conjugation of peptides *via* the cysteine sulfhydryl group to the maleimide located on the distal end of the PEG chain as carried out in the present manuscript leads to the formation of a stable thioether¹¹. The thioether linkage has low polarity and therefore is not subjected to heterolytic cleavage under normal conditions¹². Of note, some protein engineering approaches introduce thioether conjugates to enhance the stability of peptides and proteins towards proteolysis and/or chemical modification¹³⁻¹⁶. For example, oxytocin has been modified by thioether formation which led to increased plasma half-life and stability towards reducing and alkylating agents¹³. Thus, we are confident that our thioether-conjugated constructs are relatively stable. Nevertheless, we are well aware of the fact that further optimization and sophisticated quality control measures are very important and, therefore, we are currently developing techniques to investigate the *in vivo* stability of the conjugates but also the integrity of functionalized PFCs in detail.

The reviewer is right that biotin tagging is an alternative option. We have chosen this system for the scFv approach because maleimide conjugation would have required the random introduction of sulfhydryl groups into scFv potentially affecting its binding specificity for the ligand-induced binding site of activated GPbII/IIIa. Furthermore, scFv was already constructed

with a terminal biotin enabling a site-specific conjugation to avidin on the surface of PFCs without affecting its binding properties. Functionalization of PFCs with the biotin system is a methodology that we use for targeting approaches where classical chemical crosslinking would have been difficult or caused binding problems. However, for ligands like nanobodies or recombinant antibodies, we are also working on other conjugation strategies which are based on enzymatic reactions¹⁷ like the sortase system or transglutaminase reactions¹⁸.

Minor comments:

- The first sentence states that anatomical markers of diseases are absent in men. Should that read “Mankind”? CVD is more prevalent in the female population. This should be rephrased in close consultation with a clinician working in the CVD field.

Sorry for this ambiguity! As suggested by the reviewer, we have rewritten the first part of the abstract again with valuable help from our colleagues of the Clinic of Cardiology (page 2, line 2 ff):

Prediction of the transition from stable to acute coronary syndromes driven by vascular inflammation, thrombosis with subsequent microembolization and vessel occlusion leading to irreversible myocardial damage is still an unsolved problem.

References

1. Bönner, F. *et al.* Ecto-5'-nucleotidase on immune cells protects from adverse cardiac remodeling. *Circ. Res.* **113**, 301–312 (2013).
2. Bönner, F. *et al.* Multifunctional MR monitoring of the healing process after myocardial infarction. *Basic Res. Cardiol.* **109**, 430 (2014).
3. Haberkorn, S. M. *et al.* Cardiovascular magnetic resonance relaxometry predicts regional functional outcome after experimental myocardial infarction. *Circ. Cardiovasc. Imaging* **10**, e006025 (2017).
4. Jelenik, T. *et al.* Insulin resistance and vulnerability to cardiac ischemia. *Diabetes* **67**, 2695–2702 (2018).
5. Engelowski, E. *et al.* IL-23R signaling plays no role in myocardial infarction. *Sci. Rep.* **8**, 17078 (2018).
6. Alter, C., Ding, Z., Flögel, U., Scheller, J. & Schrader, J. A2bR-dependent signaling alters immune cell composition and enhances IL-6 formation in the ischemic heart. *Am. J. Physiol. Heart Circ. Physiol.* (2019) doi:10.1152/ajpheart.00029.2019.
7. Petz, A. *et al.* Cardiac hyaluronan synthesis is critically involved in the cardiac macrophage response and promotes healing after ischemia reperfusion injury. *Circ. Res.* (2019) doi:10.1161/CIRCRESAHA.118.313285.
8. Henning, C. *et al.* Endothelial β 1 integrin-mediated adaptation to myocardial ischemia. *Thromb. Haemost.* (2021) doi:10.1055/s-0040-1721505.

9. Wang, X. *et al.* Fluorine-19 magnetic resonance imaging of activated platelets. *J. Am. Heart. Assoc.* **9**, e016971 (2020).
10. Ciesiński, K. L. & Caravan, P. Molecular MRI of thrombosis. *Curr. Cardiovasc. Imaging Rep.* **4**, 77–84 (2010).
11. Hermanson, G. T. *Bioconjugate Techniques*. (Academic Press, 2013).
12. Fox, M. A. & Whitesell, J. *Organic Chemistry*. (Jones and Bartlett Publishers, 2003).
13. Muttenthaler, M. *et al.* Modulating oxytocin activity and plasma stability by disulfide bond engineering. *J. Med. Chem.* **53**, 8585–8596 (2010).
14. Gerona-Navarro, G. *et al.* Rational design of cyclic peptide modulators of the transcriptional coactivator CBP: a new class of p53 inhibitors. *J. Am. Chem. Soc.* **133**, 2040–2043 (2011).
15. Lindgren, J. & Eriksson Karlström, A. Intramolecular thioether crosslinking of therapeutic proteins to increase proteolytic stability. *Chembiochem* **15**, 2132–2138 (2014).
16. Nilsson, A., Lindgren, J. & Eriksson Karlström, A. Intramolecular thioether crosslinking to increase the proteolytic stability of affibody molecules. *Chembiochem* **18**, 2056–2062 (2017).
17. Antos, J. M. *et al.* Site-specific protein labeling via sortase-mediated transpeptidation. *Curr. Protoc. Protein Sci.* **89**, 15.3.1-15.3.19 (2017).
18. Sadiki, A. *et al.* Site-specific conjugation of native antibody. *Antib. Ther.* **3**, 271–284 (2020).

Response to Reviewer #3

The authors present an interesting preclinical study with outstanding imaging technique performed at 9.4 T. At this high field magnetic strength ¹⁹Fluorine signal, spectral resolution, and spatial resolution are very high. However, the clinical situation at 3 T, would not as efficacious.

Thank you for rating our imaging technique as outstanding – we were very happy to read this assessment! We agree that compared to 9.4 T, signal intensity, spectral and spatial resolution at 3 T will be lower, but on the other hand, compared to mice, human dimensions are substantially larger. Thus, the voxel size in cardiac MR diagnostics at 3 T is in the range of 2 to 30 μ L, whereas it was only 0.2 to 0.4 μ L in our study at 9.4 T, which translates into a substantial sensitivity increase in the clinical setting. In line with this, we and others have already shown that *in vivo* ¹⁹F MRI can robustly carried out at 3T^{19–22}. Furthermore, please take note of Supplemental Figure 2 from Reference 23: The individual signals used for mCSSI in the present study are also well resolved at 3T (for your convenience see screenshot of this figure below).

Comments:

1. While ¹⁹F background in mice and man is negligible, the prevalence of activated macrophages and smoldering inflammation throughout the human body is very high, especially on a Western Diet or with a myriad of inflammation associated disease. The uptake

of PFC particles by splenic, bone marrow, and blood phagocytic cells (Macs) is always high if saturated as in this study, and migration of activated macrophages to sites of inflammation is constantly occurring. Now seen as a snapshot. In the mouse, the early forms of atherosclerosis are induced on a high fat diet and the PFC particles serve as a phagocytosed marker to see that activity. The situation in man is more complicated.

We fully agree that atherosclerosis in mice may differ from atherosclerosis as we see it in humans. In patients we see stronger inflammatory responses that lead to plaque instability associated with microthrombi and rupture. These phenomena are typically not seen in classical mouse models of atherosclerosis. As such, in the human setting our imaging approach might be even more relevant. Along this line of comparing mouse and humans, we recently demonstrated that human monocytes are five times more capable of PFC uptake than mouse monocytes²³ indicating even enhanced sensitivity in the human setting. Importantly, these findings in healthy volunteers were also confirmed for blood monocytes obtained from patients with stable coronary artery disease or acute ST-elevation myocardial infarction²⁴. Thus, the situation in man is challenging, but also very promising.

2. The consideration of simultaneous fluorine spectral analysis is not itself new and has been discussed for the last 20 years and demonstrated by other labs long ago, mostly by implanting the PFC particles in different location, perhaps after stem cell uptake ex-vivo with two types of PFOB and PFCE.

We kindly disagree with the Reviewer here and would like to explain why we view this differently: The study by Partlow et al. (2007)²⁵ the reviewer is most likely referring to did not use simultaneous but sequential fluorine imaging, which means that the images of PFOB and PFCE were acquired consecutively. This procedure is time-consuming and prone to localization errors due to movement during the lengthy acquisition period. For this reason, we have developed in the present study multi chemical shift selective imaging (mCSSI) which allows true simultaneous acquisition of, in principle, any numbers of frequencies in one MRI scan. This has never been reported before.

Furthermore, the cited study just used simple passive *ex vivo* labelling of stem cell lines, which were subsequently implanted. In contrast, we have combined mCSSI with a multi-targeted nanotracer platform technology that allows the specific *in situ* labelling and *in vivo* visualization of multiple epitopes. This true simultaneous multiplex imaging combined with differentially targeted nanotracers allowed us to introduce an entirely new concept of ‘danger pattern’ imaging that was able to predict the site of major adverse cardiovascular events. This has the potential to develop as a ground-breaking approach for precision medicine to optimize risk

management and tailored therapy. Whether this will hold true in the clinical setting is still unknown but it clearly opens new avenues in the imaging field never been reported before.

3. The images of the three PFCs spectra are in the same location of the heart. The authors provide no data regarding the differential uptake of PFC particles through phagocytosis pathways and cell surface targeting. The general endocytic pathway may account for 80% of the signal versus targeting, despite pegylation over time. Perhaps by blocking phagocyte endocytosis with Cytochalasin D ex vivo, one might help in rooting out this question over 2 or more day time frame. From the data shown, there is no evidence that targeting matters. All the spectral overlay in the same cardiac region, and may be there by nonspecific uptake and trafficking of activated Macs from blood, spleen, or marrow.

Again, we kindly disagree and would like to suggest going together with a closer look through our crucial figures. Following the Reviewer's argument, we would have observed signals from all PFCs at the same locations. However, as seen in Figure 3, there is a change of the individual PFC patterns over time and in Figure 4 their locations are completely different. PEGylation – although not 100% perfect – is well-established to suppress phagocytic uptake of nanoparticles ensuring specificity for targeting. We have recently provided compelling evidence that this applies also for PFCs and, importantly, that PEGylation of PFCs strongly reduces the cellular uptake by monocytes and macrophages.^{26,27} Moreover, we have demonstrated that combination of PEGylation with specific targeting ligands is suitable to outcompete the phagocytic uptake by monocytes/macrophages and to specifically retarget PFCs to individual cells or thrombi.^{9,26,28,29}

Nevertheless, we take this very seriously and, thus, to further address the concerns of the Reviewer, we carried out additional uptake experiments of RAW macrophages as well as murine monocytes exposed to neat, PEGylated and PEGylated targeted PFCs, respectively (Supplementary Fig. 8, see also screenshot of the Figure below). These data unequivocally demonstrate that PEGylation indeed blunts the uptake of targeted PFCs by phagocytic cell types, thereby ensuring their specificity.

4. Thromboembolism is not uncommon with PFC particles, and the cause of the PE reported may be due to aggregation of PFC particles getting entrapped in the lungs. This is what is likely seen. The authors report sudden deaths with two mice, and that happens when particles are injected too fast, there cell surface homing chemistries interact, or other nonspecific aggregation during the injection process. It happens.

We can address this Reviewer's concern: The death of both mice was not related to the PFC injection. The mice died due to spontaneous myocardial infarction which is a unique characteristic of the used animal model.^{30,31} We have been working with PFCs for a long time and never observed any complications with proper PFC formulations. For this, we always carry out a careful quality control of the different PFC preparations in terms of size, size distribution and ζ potential and the PFC injections are performed gently and slowly. Of note, there is no evidence in the literature that thromboembolism is induced by PFCs. In line with this, we never found any indication for cytotoxicity or complement activation which might induce thrombosis or other side effects.^{26,29,32,33}

5. The dose of the PFC particles administered is very high, when considered on a human scaled use. How much of this dose is needed to saturate the macrophage targets and how much is required to overcome rodent-related hepatobiliary clearance. All rodents rapidly clear particles, even up to 300nm or more, through the biliary into the gut. Humans, do not. Some of the PFC used have very long biological half-lives and will be persistent. These doses of PFC particles have been associated adverse flu-like symptoms in man, during the days of artificial blood substitute work, and probably do in mice too. They just don't talk.

The reviewer is right, that the applied PFC dose is relatively high if you think of scaling up this to a human setting. However, PFC doses can be reduced for human use (see #1), biological half-lives can be tailored by the physicochemical properties of PFCs and the flu-like symptoms in man were related to the first generation of PFC emulsions used in the very early days of artificial blood substitute work.^{34,35} Importantly, the emulsifiers used in this study which build the scaffold of the lipid layer are derived from Lipoid and are approved for clinical use for pharmaceutical drug delivery or parenteral nutrition. Of note, PFCs are not cleared through the biliary into the gut – they are cleared *via* exhalation through the lungs in both rodents and humans.³⁴

6. Since PFC particles will persist at the inflammation sites, the serial uptake of PFC particles is the sum of signal carry-over from the previous injection plus the migration of current inflammation to the site. Like a pulse chase experiment. The residual 19F signal prior to each serial injection needs to be provided to understand the incremental changes.

We would kindly ask the Reviewer to have another and closer look at our experiments and the figures reporting our data. The requested experiment is exactly what we have done in this study (Figures 1, 3 and 4). We are happy to be able to comply so fully.

7. The uptake of particles, including PFC, may actually incite macrophages to polarize to an M1-like phenotype and this may also contribute to the migration flux to inflamed sites. Perhaps more consideration of this is warranted.

Of course, this issue is relevant for all cell targeting experiments and, thus, it has already been intensely addressed by us and others. From our data and the literature, there is no evidence for any effects of PFC uptake on the phenotype of macrophages, monocytes, T cells and dendritic cells.^{26,32,36–38}

Overall, while the study is interesting from a strict MRI community sense, the translational impact is low, detracting from the general readership. The innovation of multispectral

imaging is old and not strongly proven as a differentiating molecular imaging technique in vivo in this experimental design. Again, all of the signal goes to a well-documented location for plaque development and inflammation proportional to the degree of disease in this model, which has been shown by this and others before.

We kindly disagree with the Reviewer's general assessment and feel confident to have provided evidence above to change this view. In our point-by-point response, we have worked out (i) the specificity of our approach for different targets and (ii) the novelty of our true simultaneous multiplex imaging combined with multi-targeted nanotracers for *in situ* tracing of different epitopes. Translationally, this allowed us to introduce an entirely new concept of 'danger pattern' imaging to predict the site of major adverse cardiovascular events. Our current toolbox holds the potential to map the transition from stable to unstable coronary syndromes covering the entire cascade from local inflammation, subsequent platelet adhesion, and microembolization that finally culminates in myocardial infarction. Full encompassment of this process will allow the exact characterization of the ongoing disease state which will not only be useful for mechanistic studies in basic research but holds great promise for clinical translation towards tailored therapy in patients.

References

1. Bönner, F. *et al.* Ecto-5'-nucleotidase on immune cells protects from adverse cardiac remodeling. *Circ. Res.* **113**, 301–312 (2013).
2. Bönner, F. *et al.* Multifunctional MR monitoring of the healing process after myocardial infarction. *Basic Res. Cardiol.* **109**, 430 (2014).
3. Haberkorn, S. M. *et al.* Cardiovascular magnetic resonance relaxometry predicts regional functional outcome after experimental myocardial infarction. *Circ. Cardiovasc. Imaging* **10**, e006025 (2017).
4. Jelenik, T. *et al.* Insulin resistance and vulnerability to cardiac ischemia. *Diabetes* **67**, 2695–2702 (2018).
5. Engelowski, E. *et al.* IL-23R signaling plays no role in myocardial infarction. *Sci. Rep.* **8**, 17078 (2018).
6. Alter, C., Ding, Z., Flögel, U., Scheller, J. & Schrader, J. A2bR-dependent signaling alters immune cell composition and enhances IL-6 formation in the ischemic heart. *Am. J. Physiol. Heart Circ. Physiol.* (2019) doi:10.1152/ajpheart.00029.2019.
7. Petz, A. *et al.* Cardiac hyaluronan synthesis is critically involved in the cardiac macrophage response and promotes healing after ischemia reperfusion injury. *Circ. Res.* (2019) doi:10.1161/CIRCRESAHA.118.313285.
8. Henning, C. *et al.* Endothelial β 1 integrin-mediated adaptation to myocardial ischemia. *Thromb. Haemost.* (2021) doi:10.1055/s-0040-1721505.
9. Wang, X. *et al.* Fluorine-19 magnetic resonance imaging of activated platelets. *J. Am. Heart. Assoc.* **9**, e016971 (2020).

10. Ciesiński, K. L. & Caravan, P. Molecular MRI of thrombosis. *Curr. Cardiovasc. Imaging Rep.* **4**, 77–84 (2010).
11. Hermanson, G. T. *Bioconjugate Techniques*. (Academic Press, 2013).
12. Fox, M. A. & Whitesell, J. *Organic Chemistry*. (Jones and Bartlett Publishers, 2003).
13. Muttenthaler, M. *et al.* Modulating oxytocin activity and plasma stability by disulfide bond engineering. *J. Med. Chem.* **53**, 8585–8596 (2010).
14. Gerona-Navarro, G. *et al.* Rational design of cyclic peptide modulators of the transcriptional coactivator CBP: a new class of p53 inhibitors. *J. Am. Chem. Soc.* **133**, 2040–2043 (2011).
15. Lindgren, J. & Eriksson Karlström, A. Intramolecular thioether crosslinking of therapeutic proteins to increase proteolytic stability. *Chembiochem* **15**, 2132–2138 (2014).
16. Nilsson, A., Lindgren, J. & Eriksson Karlström, A. Intramolecular thioether crosslinking to increase the proteolytic stability of affibody molecules. *Chembiochem* **18**, 2056–2062 (2017).
17. Antos, J. M. *et al.* Site-specific protein labeling via sortase-mediated transpeptidation. *Curr. Protoc. Protein Sci.* **89**, 15.3.1-15.3.19 (2017).
18. Sadiki, A. *et al.* Site-specific conjugation of native antibody. *Antib. Ther.* **3**, 271–284 (2020).
19. Rothe, M. *et al.* In vivo ¹⁹F MR inflammation imaging after myocardial infarction in a large animal model at 3 T. *Magn. Reson. Mater. Phy.* **32**, 5–13 (2019).
20. Ahrens, E. T., Helfer, B. M., O’Hanlon, C. F. & Schirda, C. Clinical cell therapy imaging using a perfluorocarbon tracer and fluorine-19 MRI. *Magn. Reson. Med.* **72**, 1696–1701 (2014).
21. Colotti, R. *et al.* Characterization of perfluorocarbon relaxation times and their influence on the optimization of fluorine-19 MRI at 3 tesla. *Magn Reson Med* **77**, 2263–2271 (2017).
22. Darçot, E. *et al.* A characterization of ABL-101 as a potential tracer for clinical fluorine-19 MRI. *NMR Biomed.* **33**, e4212 (2020).
23. Bönner, F. *et al.* Monocyte imaging after myocardial infarction with ¹⁹F MRI at 3 T: a pilot study in explanted porcine hearts. *Eur. Heart J. Cardiovasc. Imaging* **16**, 612–620 (2015).
24. Nienhaus, F. *et al.* Phagocytosis of a PFOB-nanoemulsion for ¹⁹F magnetic resonance imaging: First results in monocytes of patients with stable coronary artery disease and ST-elevation myocardial infarction. *Molecules* **24**, (2019).
25. Partlow, K. C. *et al.* ¹⁹F magnetic resonance imaging for stem/progenitor cell tracking with multiple unique perfluorocarbon nanobeacons. *FASEB J.* **21**, 1647–1654 (2007).
26. Temme, S. *et al.* Synthetic cargo internalization receptor system for nanoparticle tracking of individual cell populations by fluorine magnetic resonance imaging. *ACS Nano* **12**, 11178–11192 (2018).
27. Bouvain, P. *et al.* Dissociation of ¹⁹F and fluorescence signal upon cellular uptake of dual-contrast perfluorocarbon nanoemulsions. *Magn. Reson. Mater. Phy.* **32**, 133–145 (2019).
28. Grapentin, C. *et al.* Optimization of perfluorocarbon nanoemulsions for molecular imaging by ¹⁹F MRI. in *Nanomedicine* (eds. Seifalín, A., de Mel, A. & Kalaskar, D. M.) 268–86 (One Central Press, 2014).
29. Temme, S. *et al.* Noninvasive imaging of early venous thrombosis by ¹⁹F magnetic resonance imaging with targeted perfluorocarbon nanoemulsions. *Circulation* **131**, 1405–1414 (2015).

30. Hermann, S. *et al.* Imaging reveals the connection between spontaneous coronary plaque ruptures, atherothrombosis, and myocardial infarctions in hypoE/SRBI^{-/-} mice. *J. Nucl. Med.* **57**, 1420–1427 (2016).
31. Zhang, S. *et al.* Diet-induced occlusive coronary atherosclerosis, myocardial infarction, cardiac dysfunction, and premature death in scavenger receptor class B type I-deficient, hypomorphic apolipoprotein ER61 mice. *Circulation* **111**, 3457–3464 (2005).
32. Flögel, U. *et al.* In vivo monitoring of inflammation after cardiac and cerebral ischemia by fluorine magnetic resonance imaging. *Circulation* **118**, 140–148 (2008).
33. Staal, A. H. J. *et al.* In vivo clearance of ¹⁹F MRI imaging nanocarriers is strongly influenced by nanoparticle ultrastructure. *Biomaterials* **261**, 120307 (2020).
34. Krafft, M. P. & Riess, J. G. Chemistry, physical chemistry, and uses of molecular fluorocarbon--hydrocarbon diblocks, triblocks, and related compounds--unique 'apolar' components for self-assembled colloid and interface engineering. *Chem. Rev.* **109**, 1714–1792 (2009).
35. Keipert, P. E. Perflubron emulsion (Oxygent(tm)): a temporary intravenous oxygen carrier. *Anesthesiol. Intensivmed. Notfallmed. Schmerzther.* **36 Suppl 2**, S104-106 (2001).
36. Ahrens, E. T., Flores, R., Xu, H. & Morel, P. A. In vivo imaging platform for tracking immunotherapeutic cells. *Nat. Biotechnol.* **23**, 983–987 (2005).
37. Srinivas, M., Morel, P. A., Ernst, L. A., Laidlaw, D. H. & Ahrens, E. T. Fluorine-19 MRI for visualization and quantification of cell migration in a diabetes model. *Magn. Reson. Med.* **58**, 725–734 (2007).
38. Srinivas, M. *et al.* In vivo cytometry of antigen-specific t cells using ¹⁹F MRI. *Magn. Reson. Med.* **62**, 747–753 (2009).

REVIEWERS' COMMENTS

Reviewer #3 (Remarks to the Author):

I am informed on some points and agree to disagree on others. That said there are two points that are significant with regard to translation that should be addressed.

The first is the issue about liver signal being due to RES uptake. Only a minor proportion of the particles are taken up by Kupfer cells in the liver, the vast majority of the particles are in the hepatobiliary system in transit to the gut. This is known for particles in rodents since the 1950's if not before and the topic was partially reviewed and illustrated by MRI in Rats as a cholangiogram. Bulte et al. This does not occur in humans or rabbits and most non rodent species. Consequently, blood clearance and particle pharmacokinetics will be greatly affected, which impacts imaging. This is magnified with high doses. Also, the doses of PFC emulsions lead to tension of the engorged liver against its encasing fascia. The consequence is physical compression of hepatocytes leading to increased cell damage and increased LFTs.

The second issue is that these particles are pegylated, which is a significant concern to the FDA and others. Pre-existing antibodies, often IgE, against PEG exist in many people leading to acute immune responses. These antibodies exist because PEG is in many products used by people, notably cosmetics. Preclinical models, regardless of species do not have this exposure and their acute exposure to the pegylated particles has no impact. Recently Definity, and microbubble echo contrast has received an FDA warning due to PEGylation. Other clinical trials have been halted when the drug being delivered, such as doxorubicin, does not simultaneously suppress immune response.

Thus while the research is interesting if not truly innovative, it is likely not translational.

Reviewer #4 (Remarks to the Author):

The authors adequately revised the manuscript.

Reviewer #5 (Remarks to the Author):

I did not review the first submission of this manuscript.

I enjoyed reading the manuscript. It is quite a technical achievement and the images are superb. In my opinion the authors have addressed the prior reviewer's concerns (I was asked to consider Reviewer 2). I have one minor, easily addressable, but important comment. The labeling of thrombus as "acute" or "advanced" is confusing and is also not consistent with the mechanism of action of the probes. All the thrombi in these experiments are acute - a day old thrombus is still acute. I don't know what they mean by "advanced"; I assume this has something to do with the age of the thrombus. The natural history of thrombosis is very well established. Fibrin (contrary to Rev 2's comment) is present immediately in a newly formed thrombus as are activated platelets. With time the activated platelets diminish and after a long period of time the thrombus can become endothelialized and fibrin is diminished. The authors have a Factor XIII specific probe and a fibrin specific probe. Better that they label the images "Factor XIII" and "Fibrin" which would be an accurate representation of what the images represent biologically. They observe signal in hyperacute thrombi with the Factor XIII probe BUT ALSO with the fibrin probe (as expected) but in somewhat older thrombi the Factor XIII signal is lost but the fibrin signal remains. Again consistent with known biology. Labeling the images and discussing them in this way would remove the confusion around thrombus age, but importantly it would resolve the confusion that the fibrin probe only recognizes older thrombi when it fact it recognizes very fresh thrombi as well. For example PET probes based on the same fibrin-targeting peptide where used to image thrombi that were only an hour old following arterial injury (see e.g. Blasi J Nucl Med. 2014;55:1157-1163). I am not suggesting that the authors do not know/understand this, rather that the manuscript as written was somewhat confusing in this regard.

RESPONSE LETTER

We thank all reviewers for the careful evaluation of our manuscript and their constructive criticism. Their suggestions and comments have helped to significantly improve our manuscript. We have addressed the various issues point by point as outlined below.

Response to Reviewer #3

I am informed on some points and agree to disagree on others.

We greatly appreciate this fair competitive scientific spirit!

That said there are two points that are significant with regard to translation that should be addressed.

The first is the issue about liver signal being due to RES uptake. Only a minor proportion of the particles are taken up by Kupfer cells in the liver, the vast majority of the particles are in the hepatobiliary system in transit to the gut. This is known for particles in rodents since the 1950's if not before and the topic was partially reviewed and illustrated by MRI in Rats as a cholangiogram. Bulte et al. This does not occur in humans or rabbits and most non rodent species. Consequently, blood clearance and particle pharmacokinetics will be greatly affected, which impacts imaging, this is magnified with high doses. Also, the doses of PFC emulsions lead to tension of the engorged liver against its encasing fascia. The consequence is physical compression of hepatocytes leading to increased cell damage and increased LFTs.

Thank you for making us aware of the paper by Bulte et al. on MR cholangiography which indeed presents surprising but also singular findings on the biodistribution of PFC nanoparticles in two individual rats. As pointed out above by the reviewer, this does not occur in humans or rabbits and most non-rodent species. Along the same line, we as well did not observe any ^{19}F signal in the bile or intestine in mice and as a consequence also not in feces¹. Thus, the relevance of this case report is difficult to interpret, since in the majority of studies the predominant route of PFC clearance is consistent with the previously described exhalation through the lungs². However, we fully agree with this reviewer that PFC formulations, doses and biological half-lives have to be specifically tailored for human use^{2,3}, but this is clearly beyond the scope of the present study and has to be adjusted in clinical trials.

With regard to (liver) toxicity: Up to now, we observed no adverse effects on animals injected with PFCs, and serum markers of liver function as well as liver histology were comparable to those of saline-treated animals^{1,4}. Similarly, also intravenous application in clinical trials –

even with the first generation PFCs – demonstrated no detrimental effects on hemodynamics, heart, liver, kidney, or hematologic functions^{5,6}. However, to address the concerns of the reviewer, we now mentioned the alternative excretion pathway (page 10, line 7 f) and clearly state in the discussion that additional studies on the safety profile of PFCs are required (page 9, line 21 ff).

The second issue is that these particles are pegylated, which is significant concern to the FDA and others. Pre-existing antibodies, often IgE, against PEG exist in many people leading to acute immune responses. These antibodies exist because PEG is in many products used by people, notably cosmetics. Preclinical models, regardless of species do not have this exposure and their acute exposure to the pegylated particles have no impact. Recently Definity, and microbubble echo contrast has received an FDA warning due to PEGylation. Other clinical trials have been halted when the drug being delivered, such as doxorubicin, does not simultaneously suppress immune response.

We agree with this reviewer that anti-PEG antibodies in patients could be a problem for administration of PEGylated drugs and nanomaterials. It has already been shown in the early 80s that PEG is immunogenic and that administration of PEGylated proteins can provoke the formation of anti-PEG antibodies⁷. Clearly, it is also of importance to take this into consideration when using PEGylated contrast agents. On the other hand, PEG is frequently incorporated into drug formulations to improve therapeutic efficacy and in 2020, there were 21 PEGylated drugs approved by the FDA and over 20 others in clinical trials⁸. This issue could be further addressed by screening for the presence of anti-PEG antibodies prior contrast agent application. On the other hand, the search for PEG alternatives has become a hot topic. Much effort is currently undertaken to explore the capacity of synthetic polymers, natural polymers, and zwitterionic polymers in nanoparticle modification towards the development of innovative and effective drug delivery systems^{8,9}. Furthermore, polyaminoacids, polyacrylamides, and, polysaccharides or CD47-derived mimicry peptides¹⁰ are promising candidates with comparable or superior properties to PEG⁹. In order to take the concerns of this reviewer into account, we now clearly address PEG immunogenicity in the discussion (page 9, line 26 ff) and point out that future studies are needed to show that the PFC modifications required for targeting (PEGylation, ligand equipment) do not result in undesired side effects (page 9, line 21 ff).

Thus while the research is interesting if not truly innovative, it is likely not translational.

We feel that the translational aspect can only be addressed in future studies. Our current work provides proof-of-concept evidence that multi-targeted ¹⁹F MRI is useful to unmask specific

danger patterns for emerging cardiovascular disease in mice. Large animal and initial human studies confirm that ^{19}F MRI will also be feasible in the clinical setting, thus underling its translational potential^{11,12}. Nevertheless, we fully agree with his reviewer that further investigations and clinical trials are required whether the present approach will also work for human applications.

To address the overall concerns of this reviewer, we now make clear that translation of the presented approach into the clinical setting critically depends on the safety profile of the developed PFCs, which has to be proved by additional studies in the future (page 9, line 21 ff, page 10, line 21 ff).

Response to Reviewer #4

The authors adequately revised the manuscript.

No response required

Response to Reviewer #5

I enjoyed reading the manuscript. It is quite a technical achievement and the images are superb. In my opinion the authors have addressed the prior reviewer's concerns (I was asked to consider Reviewer 2).

Thank you for this positive evaluation!

I have one minor, easily addressable, but important comment. The labeling of thrombus as "acute" or "advanced" is confusing and is also not consistent with the mechanism of action of the probes. All the thrombi in these experiments are acute - a day old thrombus is still acute. I don't know what they mean by "advanced"; I assume this has something to do with the age of the thrombus. The natural history of thrombosis is very well established. Fibrin (contrary to Rev 2's comment) is present immediately in a newly formed thrombus as are activated platelets. With time the activated platelets diminish and after a long period of time the thrombus can become endothelialized and fibrin is diminished. The authors have a Factor XIII specific probe and a fibrin specific probe.

Better that they label the images "Factor XIII" and "Fibrin" which would be an accurate representation of what the images represent biologically. They observe signal in hyperacute thrombi with the Factor XIII probe BUT ALSO with the fibrin probe (as expected) but in somewhat older thrombi the Factor XIII signal is lost but the fibrin signal remains. Again consistent with known biology. Labeling the images and discussing them in this way would remove the confusion around thrombus age, but importantly it would resolve the confusion that the fibrin probe only recognizes older thrombi when it fact it recognizes very fresh thrombi as well. For example PET probes based on the same fibrin-targeting peptide where used to image thrombi that were only an hour old following arterial injury (see e.g. Blasi J Nucl Med. 2014;55:1157-1163). I am not suggesting that the authors do not know/understand this, rather that the manuscript as written was somewhat confusing in this regard.

Thank you for making us aware of these inconsistencies. We agree and have altered the labeling of the images and the wording in the manuscript as suggested by this reviewer (Figures 3+4, corresponding legends and places in the text).

References

1. Staal, A. H. J. *et al.* In vivo clearance of 19F MRI imaging nanocarriers is strongly influenced by nanoparticle ultrastructure. *Biomaterials* **261**, 120307 (2020).
2. Krafft, M. P. & Riess, J. G. Chemistry, physical chemistry, and uses of molecular fluorocarbon--hydrocarbon diblocks, triblocks, and related compounds--unique 'apolar' components for self-assembled colloid and interface engineering. *Chem. Rev.* **109**, 1714–1792 (2009).
3. Keipert, P. E. Perflubron emulsion (Oxygent(tm)): a temporary intravenous oxygen carrier. *Anesthesiol. Intensivmed. Notfallmed. Schmerzther.* **36 Suppl 2**, S104-106 (2001).
4. Flögel, U. *et al.* In vivo monitoring of inflammation after cardiac and cerebral ischemia by fluorine magnetic resonance imaging. *Circulation* **118**, 140–148 (2008).
5. Lane, T. A. Perfluorochemical-based artificial oxygen carrying red cell substitutes. *Transfus. Sci.* **16**, 19–31 (1995).
6. Castro, C. I. & Briceno, J. C. Perfluorocarbon-based oxygen carriers: Review of products and trials. *Artif. Organs* **34**, 622–634 (2010).
7. Richter, A. W. & Akerblom, E. Antibodies against polyethylene glycol produced in animals by immunization with monomethoxy polyethylene glycol modified proteins. *Int. Arch. Allergy Appl. Immunol.* **70**, 124–131 (1983).
8. Freire Haddad, H., Burke, J. A., Scott, E. A. & Ameer, G. A. Clinical relevance of pre-existing and treatment-induced anti-poly(ethylene glycol) antibodies. *Regen. Eng. Transl. Med.* 1–11 (2021) doi:10.1007/s40883-021-00198-y.
9. Hoang Thi, T. T. *et al.* The importance of poly(ethylene glycol) alternatives for overcoming PEG immunogenicity in drug delivery and bioconjugation. *Polymers (Basel)* **12**, (2020).
10. Gheibihayat, S. M., Jaafari, M. R., Hatamipour, M. & Sahebkar, A. Improvement of the pharmacokinetic characteristics of liposomal doxorubicin using CD47 biomimickry. *J Pharm Pharmacol* **73**, 169–177 (2021).
11. Ahrens, E. T., Helfer, B. M., O'Hanlon, C. F. & Schirda, C. Clinical cell therapy imaging using a perfluorocarbon tracer and fluorine-19 MRI. *Magn. Reson. Med.* **72**, 1696–1701 (2014).
12. Rothe, M. *et al.* In vivo 19F MR inflammation imaging after myocardial infarction in a large animal model at 3 T. *Magn. Reson. Mater. Phy.* **32**, 5–13 (2019).